# LLM-CXR: Instruction-Finetuned LLM for CXR Image Understanding and Generation

**Suhyeon Lee**[*], **Won Jun Kim**[*], **Jinho Chang & Jong Chul Ye**
Korea Advanced Institute of Science & Technology,
{suhyeon.lee, wonjun, jinhojsk515, jong.ye}@kaist.ac.kr

## Abstract

Following the impressive development of LLMs, vision-language alignment in LLMs is actively being researched to enable multimodal reasoning and visual input/output. This direction of research is particularly relevant to medical imaging because accurate medical image analysis and generation consist of reasoning based on a combination of visual features and prior knowledge. Many recent works have focused on training adapter networks that serve as an information bridge between image processing (encoding or generating) networks and LLMs; but presumably, in order to achieve maximum reasoning potential of LLMs on visual information as well as language, image and text features should be allowed to interact more freely. This is especially important in the medical domain because understanding and generating medical images such as chest X-rays (CXR) require not only accurate visual and language-based reasoning but also a more intimate mapping between the two modalities. Thus, taking inspiration from previous work on the transformer and VQ-GAN combination for bidirectional image and text generation, we build upon this approach and develop a method for instruction-tuning an LLM pre-trained only on text to gain vision-language capabilities for medical images. Specifically, we leverage a pretrained LLM's existing question-answering and instruction-following abilities to teach it to understand visual inputs by instructing it to answer questions about image inputs and, symmetrically, output both text and image responses appropriate to a given query by tuning the LLM with diverse tasks that encompass image-based text-generation and text-based image-generation. We show that our model, LLM-CXR, trained in this approach shows better image-text alignment in both CXR understanding and generation tasks while being smaller in size compared to previously developed models that perform a narrower range of tasks.

## 1 Introduction

The last few years have seen remarkable development in the field of Large language models (LLMs). LLMs are considered a different class of AI models because of their ability to flexibly understand/generate natural language and perform language-based reasoning, allowing them to generalize to a variety of given tasks without the need to be explicitly trained for them. As a next step, methods to enable the input of visual information alongside language in LLMs (OpenAI, 2023; Liu et al., 2023; Alayrac et al., 2022; Li et al., 2023) as well as methods that output images from LLMs (Koh et al., 2023a;b) are being actively developed. These models have great potential to be particularly useful in the medical domain, as working with medical images such as chest X-rays (CXRs) requires the ability to understand context, perform reasoning, and communicate conclusions in both image and text forms. The first generation of medical multimodal LLMs has begun to emerge recently (Moor et al., 2023; Thawkar et al., 2023; Xu et al., 2023a).

The main challenge in developing these models is achieving alignment between the pretrained language features of LLMs and the newly introduced image features without catastrophic forgetting of their language abilities. This is a more difficult challenge in the domain of medical images compared to natural images because the model needs to distinguish subtle differences in images or even parts of images

---

[*]These authors contributed equally to this work.

[†]In order to comply with the MIMIC-CXR data usage license (Johnson et al., 2019a), all CXR images presented in the Figures 2, 3, 4, 8 are replaced with similar CXR's from the Indiana University chest X-ray dataset (Demner-Fushman et al., 2016); and the presented MIMIC text reports are paraphrased.

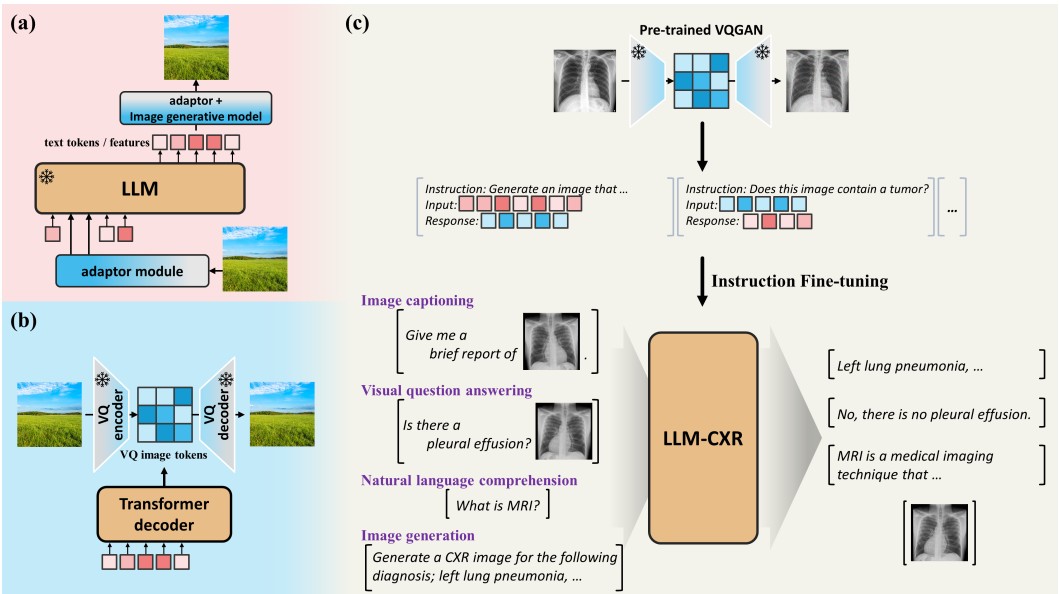

Figure 1: **(a)** Example of previous work that indirectly implements multimodal bidirectional LLM by connecting a pretrained image encoder or image generation model to a pretrained LLM with a mapping layer. **(b)** Example of previous work that implements multimodal bidirectional non-LLM transformer with VQ-GAN trained from scratch (*i.e.,* without learned language features). **(c)** To enable direct multimodal feature interaction in LLMs pre-trained with text, our method implements (b) through LLM-specific instruction fine-tuning scheme.

(*e.g.*, pneumonia vs. pulmonary edema on CXR) and then provide accurate text descriptions or image generations. Natural images tend to be more diverse than medical images, and each one can be described by a broad range of statements. Medical images, on the other hand, require highly specific yet comprehensive descriptions - making the space for correct answers much smaller and more complex. This means that a medical multimodal LLM requires a more intimate mapping between textual and visual features.

Currently, the most popular approach to map visual features to and from an LLM is to train an 'adapter network' to act as a mapping layer that translates the output of a pretrained image encoder network to a form that can be understood by an LLM (Alayrac et al., 2022; Li et al., 2023; Zhu et al., 2023) or to connect the output of an LLM to an image-generating network to output images (Koh et al., 2023a;b). In these approaches, LLMs are frozen to prevent forgetting their language and reasoning capabilities. These multimodal LLMs have demonstrated impressive capabilities in vision-language tasks such as image captioning, zero-shot classification, visual question and answering (VQA), image generation, and image retrieval. However, with this common approach of using adapter networks, vision-language alignment may be limited as the adapter network serves as an information bottleneck that can hinder the interplay between visual and language features.

To better bridge the gap between image and text, we take inspiration from the field of vision-language pertaining (VLP) with non-LLM transformers, where there has already been a lot of work on treating images and text in the same token embedding space. Most prominent is the approach that tokenizes images using VQ-GAN (Esser et al., 2021) (VQ-VAE (Van Den Oord et al., 2017)) and generates sequences of both text tokens and image tokens using an autoregressive transformer decoder (Zhang et al., 2021; Lu et al., 2022; Wang et al., 2022a; Lee et al., 2023). These previous works fit well with our view that for better image-text alignment, models should be able to process images and text equally without a separate adapter bottleneck. Moreover, the fine details of the CXR images such as texture are important in medical diagnosis, making the tokens from local image features from VQ encodings preferable targets for alignment than the global descriptions. Hence, in this work, we take advantage of the widely used architectural component VQ-GAN to seamlessly integrate the image-text token space without requiring any structural modifications to the underlying base LLM.

Building on this foundation, we propose a method for achieving better image-text alignment in LLMs for CXR image understanding and generation by leveraging an LLM's built-in instruction-following abilities. Specifically, we seek to teach the model visual information by giving it diverse instructions surrounding

CXR image analysis and generation and then using the outputs to finetune the LLM. As such, one of our main contributions is the development of this instruction-finetuning method that has been tailored to be suitable for an already-trained LLM to expand its capabilities to input and output images (tokenized by VQ-GAN) without modification of model structure or objectives. An important distinction from previous work such as (Zhang et al., 2021; Lu et al., 2022; Wang et al., 2022a; Lee et al., 2023) is that while non-LLM VLP transformers were trained from scratch without a previous understanding of language - meaning there was no concern of forgetting nor the opportunity to take advantage of its language understanding - our contribution is a method that takes a pretrained LLM and adds bidirectional multimodal capabilities by a simple instruction-finetuning process designed for LLMs (further detailed in Section 2). Through this novel approach, we produce a finetuned LLM proficient in bidirectional, multimodal tasks capable of CXR-to-report generation, report-to-CXR generation, and CXR-related VQA. We show that this model has state-of-the-art image-text understanding and generative capabilities by demonstrating that it outperforms previously developed models in each of these tasks even though the other models were specifically designed for only a subset of the tasks.

## 2 LLM-CXR

### 2.1 CLINICAL INFORMATION-PRESERVING CXR TOKENIZATION

For the tokenization process of the images, we used VQ-GAN (Esser et al., 2021), a widely used approach (Wang et al., 2022a; Lu et al., 2022; Zhang et al., 2021; Lee et al., 2023) for tokenizing images in multimodal generation models with transformers. More specifically, we utilize the quantized latent space of the VQ-GAN model trained on the image domain. VQ-GAN consists of a frozen encoder $E(\cdot) : \mathbb{R}^{C \times H \times W} \to \{1, 2, ..., K_{img}\}^{d_z}$, decoder $D(\cdot) : \{1, 2, ..., K_{img}\}^{d_z} \to \mathbb{R}^{C \times H \times W}$, and codebook $C \in \mathbb{R}^{K_{img} \times n_z}$ that contains $K_{img}$ codes. With this VQ-GAN, it is possible to obtain tokenized image $z \in \{1, 2, ..., K_{img}\}^{d_z}$ of length $d_z$. As shown in Figure 1(c), this allows us to freely convert images into tokens and then back to images similar to an autoencoder (Kramer, 1991). Furthermore, the tokenized images contain more localized information in each token, making them suitable for medical diagnosis purposes where localized and texture information is also critical. The VQ-GAN remains frozen during the training of the LLM. Its sole purpose is to encode and decode images, facilitating their input to and output from the LLM similar to tokenizers for text. Consequently, the LLM operates with images in the form of these image tokens, both for input and output processes.

However, the original VQ-GAN's reconstruction objective during training only consists of L1 loss and LPIPS loss (Zhang et al., 2018), which causes loss of clinically important information such as characteristics of microscopic lesions in the information bottleneck formed by the quantization process. Therefore, to minimize the loss of such important but subtle information in CXRs, we present another contribution: an additional 1024-dimensional feature L2 reconstruction loss extracted from the CXR encoder model of the TorchXRayVision (Cohen et al., 2022) library that is used when training the VQ-GAN for image tokenization. This *clinical information-preserving CXR tokenization* leads to performance improvement in both the report-to-CXR task and the CXR-to-report task.

### 2.2 EXPANDING LLM'S TOKEN EMBEDDING SPACE

Non-LLM transformers are trained with both images and text from the beginning. Training multimodal LLMs has a subtle but crucial difference: LLMs are already trained on text, and this text-based training is too expensive to be done again during multimodal training. Thus, the goal is to confer visual capabilities using a fine-tuning process to an LLM previously trained only on the text in a way that the newly introduced visual information is in line with the pre-existing language information. To place image tokens in the same embedding space as text tokens without losing the language abilities of the LLM, we treated the process of adding image tokens to the model for fine-tuning the same as the technique of increasing the special token in the vocabulary (*i.e.*, token type) in language model finetuning for the image retrieval and generation (Koh et al., 2023b;a). Concretely, if the LLM's original embedding table was $\mathbb{R}^{K_{text} \times d_e}$ in which the embedding dimension is $d_e$, then the embedding table is expanded to $\mathbb{R}^{(K_{text} + K_{img}) \times d_e}$. The existing elements are retained and used as initial values for fine-tuning, while the newly expanded parts are initialized randomly. The entire embedding table is trainable during the fine-tuning process.

## 2.3 DATA AUGMENTATION WITH SYNTHETIC VQA

Text reports for CXRs contain a comprehensive, detailed description of the CXR image in question. While these image-text pairs can be used as-is to achieve vision-language alignment, this training can be enhanced by taking the text report and generating visual questions and answers (VQAs) that can be asked about CXR images. This is not only a way to further enhance vision-language alignment but also an important way to ensure that the natural language interaction capability is maintained in the model. We show that utilizing this multi-task instruction-tuning approach improves performance on all fronts (VQA, text-based image generation, and image-based text generation).

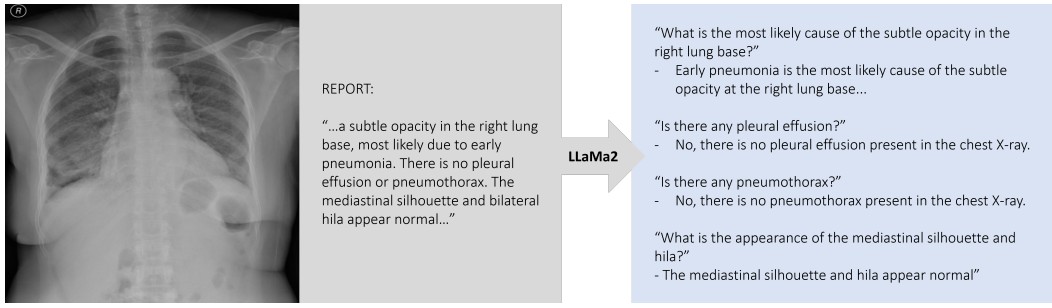

Figure 2: Examples of VQA generated from a CXR text report.[†]

We use LlaMa 2 (`Llama2-13b-chat-hf`) (Touvron et al., 2023) to generate questions and answers about a chest X-ray as shown in Figure 2. Specifically, about ∼200,000 CXRs that were labeled in the MIMIC-CXR-JPG dataset to be positive for one or more lesions were selected, and LlaMa 2 was prompted to generate a few questions for each CXR. The prompt used to generate these VQAs and more examples of generated VQAs are included in Appendix B.

## 2.4 IMAGE-TEXT BIDIRECTIONAL INSTRUCTION FINE-TUNING

Taking inspiration from previous methods for non-LLM transformers' multimodal generative methods (Wang et al., 2022a; Lu et al., 2022; Zhang et al., 2021; Lee et al., 2023), we adopt and transform this technique into an instruction finetuning (Wang et al., 2022b; Wei et al., 2021) scheme suitable for LLMs pretrained on a large text corpus. Since this process is simply a fine-tuning process for LLM, no structural or objective changes are made to LLM other than the expansion of the token embedding table, and no additional networks are required. The template for instruction-finetuning uses the template used by the Alpaca family (Taori et al., 2023; Databricks, 2023). Appendix C is the template of the prompt for instruction fine-tuning from the Alpaca family which consists of *Instruction*, *Input*, *Response* sections.

During the fine-tuning process, the LLM is optimized according to the objective function that outputs a response based on the instruction-input pairs in an autoregressive manner. Note that this is an *instruction-tuning* scheme, inheriting but distinct from the training of non-LLM transformer multimodal generation methodologies. The advantage of this scheme will be covered in more detail later in Section 2.4.1.

The tasks used for fine-tuning are categorized into four main types: 1) natural language instruction-following (NL-IF), 2) report-to-CXR generation, 3) CXR-to-report generation, and 4) CXR-based vision question answering (CXR-VQA). These are the four primary task types categorized based on input and output modalities; but for the model, they form a rich training environment with a wide spectrum of tasks that are distinguished by *Instruction*. NL-IF and CXR-VQA training examples provide multi-dimensional tasks so that the model can learn intricately aligned visual and textual features and generalization of tasks instructed in natural language. The report-to-CXR and CXR-to-report generation tasks are used in high volume during training and are important for vision-language alignment, but it must be noted that they are merely two tasks among many. Through the use of *Instructions* to specify tasks, we *add* several unseen multimodal task capabilities to the base LLM without overwriting existing language-based interactive capabilities. It also enables simpler yet more general user interaction compared to the existing non-LLM multimodal bidirectional generation models discussed above, as they can only be queried for certain tasks using predefined tokens, while LLM-based models enable queries based on natural language instructions and thus the possibility of generalizing to zero-shot tasks (Wei et al., 2021; Wang et al., 2022b).

**NL-IF task.** We initialize the base LLM with weights of a pretrained instruction-following LLM (Databricks, 2023). To minimize the risk of forgetting of language proficiency during the fine-tuning

process, we concurrently engage in instruction-following tuning using the same NL-IF dataset used to instruction-tune this base LLM.

**Report-to-CXR generation.** This is a task that aims to generate CXR images that match the *Input* radiology report as a *Response*. The *Instructions* for this task are randomly sampled from 10 versions similar to the instructions in the example below. The LLM directly outputs image tokens in the same way as text tokens due to the utilization of the expanded token space encompassing both text and image tokens. Therefore, CXR image generation does not require an additional network or text-to-image generative model (e.g. stable diffusion) as seen in Wu et al. (2023b); Koh et al. (2023a). Below is an example instruction/input pair used to instruct the LLM to generate a CXR image.

```
### Instruction: Generate a chest X-ray image that corresponds
     to the entered free-text radiology reports for the chest X-ray image.
Input: Bilateral, diffuse, confluent pulmonary opacities. Differential
     diagnoses include severe pulmonary edema ARDS or hemorrhage.
### Response: <VQ032 VQ015 VQ124 ... VQ054 VQ032>
```

**CXR-to-report generation.** In this task, tokenized CXR images are the *Input*. Note that in our model, the image is not processed through a separate network trained with paired vision-language data as in Li et al. (2023); Liu et al. (2023); the tokenized image is directly into the LLM, and the LLM itself learns visual information on top of its language capabilities. It can then be instructed to output a corresponding radiology report to a given image as *Response*. *Instruction*s are also randomly sampled from ten versions similar to the example below. The following snippet is an example instruction for the CXR-to-report task.

```
### Instruction: Generate radiology reports for the entered CXR image.
Input: <VQ071 VQ057 VQ 402 ... VQ122 VQ002>
### Response: No acute cardiopulmonary process.
```

**CXR-VQA task.** In this task, questions about an image are given as *Instructions*, and the model generates an appropriate *Response*. Questions are about the CXR images given as *Input*. This not only trains the model to gain VQA capabilities but also improves the performance in the other vision-language tasks as well (*i.e.*, enhancement of vision-language alignment). Below is an example instruction for the CXR-VQA task.

```
### Instruction: What is the size of the pleural effusions?
Input: <VQ121 VQ720 VQ002 ... VQ005 VQ428>
### Response: The bilateral pleural effusions are moderate to large.
```

### 2.4.1 TRAINING OBJECTIVE

The training objective is to generate the entire target paragraph which consists of *Instruction*, *Input*, and *Response* in an autoregressive manner. However, similar to general GPT pre-training (Radford et al., 2018; 2019; Brown et al., 2020), the loss is only applied to the tokens generated after the response key (*i.e.*. `### Response:`), following an instruction-tuning scheme (Taori et al., 2023; Databricks, 2023).

Specifically, for tokenized training paragraph $[x_1, x_2, ..., x_{n_x}, y_1, y_2, ..., y_{n_y}]$ where $\boldsymbol{x}$ denotes *Instruction* and *Input* sections and $\boldsymbol{y}$ denotes *Response* section, the training loss is given by:

$$L_{instruct} = -\log p(\boldsymbol{y}|\boldsymbol{x}) = \sum_{i=1}^{n_y} -\log p(y_i|y_{i-1}, y_{i-2}, ..., y_1, x_{n_x}, x_{n_x-1}, ..., x_1). \tag{1}$$

Note that this objective is different from the one used to train non-LLM transformers such as in Lee et al. (2023) which uses $L_{joint} = -\log p(\boldsymbol{x}, \boldsymbol{y})$. We hypothesize that instruction-tuning through the use of conditional loss mitigates overfitting to the fine-tuning dataset, particularly when working with limited data, thus promoting a better learning environment that encourages the model to expand its understanding from its pre-existing features rather than memorizing new features.

### 2.4.2 TWO-STAGE FINE-TUNING

Similar to previous works Chambon et al. (2022); Xu et al. (2023b), we place our focus on frontal view (*i.e.*, AP and PA) images and the *Impression* sections of corresponding reports as they are the most relevant to making diagnoses and more amenable to straightforward comparison of the results. Additionally, we minimize references to prior studies within reports (as we will use just one image at a time during inference

and it would not make sense to have reports that refer to prior studies) by pruning the MIMIC-CXR-JPG dataset to only include the first study for each subject.

Drawing inspiration from multi-stage training techniques commonly used in recent LLM fine-tuning methods (Zhu et al., 2023), we follow a similar two-stage training approach. In the first stage, unfiltered high-volume data (i.e., all CXR image-report pairs and findings/impression sections in the MIMIC-CXR dataset) is used to train the model on the entire distribution of new image tokens and the general relationship between image and text tokens. In the second stage, model performance is further enhanced using the pruned dataset (i.e., using only frontal view images, the first study for each patient, and impression sections of text reports) that provides a stronger signal for vision-language alignment.

Implementation details are provided in the Appendix A.

## 3 EXPERIMENTS

We compare the performance of LLM-CXR against similar contemporary models across all tasks performed by LLM-CXR. For CXR-to-report generation, we compare our results with UniXGen (Lee et al., 2023), XrayGPT (Thawkar et al., 2023), RadFM Wu et al. (2023a), IFCC Delbrouck et al. (2022) and R2Gen Chen et al. (2020); for CXR-VQA, with XrayGPT (Thawkar et al., 2023), RadFM Wu et al. (2023a), and ELIXR (Xu et al., 2023a); and for text-to-CXR generation, with UniXGen (Lee et al., 2023) and RoentGen (Chambon et al., 2022). Note that only LLM-CXR is capable of performing all three tasks. For further details regarding the ablation experiments on the design of our method, please refer to Appendix E.

### 3.1 CXR-TO-REPORT GENERATION TASK

We use each model to generate text radiology reports for CXR images in the MIMIC-CXR-JPG dataset using LLM-CXR (Figure 3), UniXGen (Lee et al., 2023), XrayGPT (Thawkar et al., 2023), RadFM Wu et al. (2023a), IFCC Delbrouck et al. (2022), and R2Gen Chen et al. (2020). In order to quantify how well the generated reports reflect the clinically significant radiologic information in the CXR images, we use the CheXpert-labeler (Irvin et al., 2019), which is a rule-based natural language processing tool that reads a text report for a CXR and extracts whether the report mentions the presence or absence of significant radiologic findings (e.g., edema, pleural effusion, etc.), and compare the extracted labels with that of ground-truth reports.

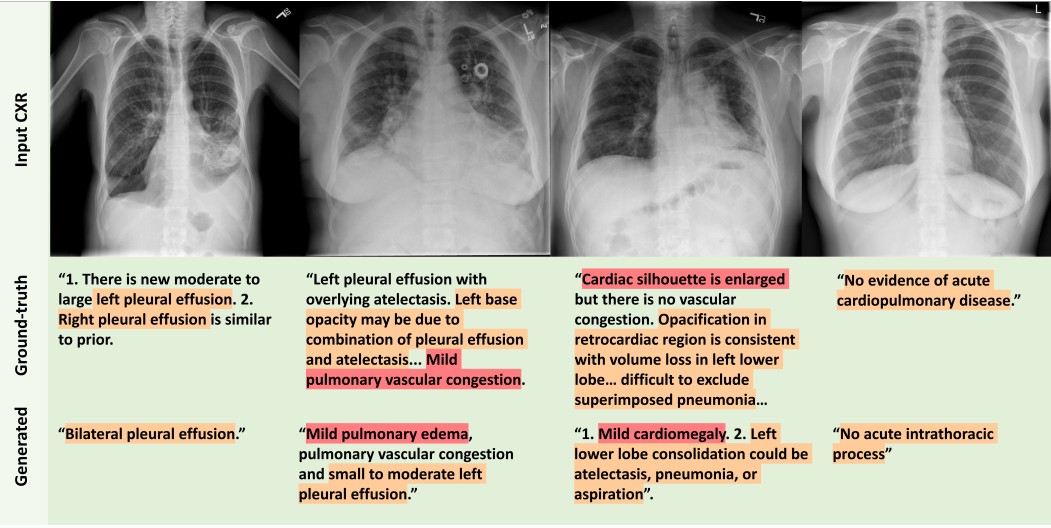

Figure 3: Examples of text report generation for a given CXR image with LLM-CXR. While the generated reports use different wording from the ground-truth reports, LLM-CXR is able to generate reports that capture the gist of the contents of the CXR, demonstrating alignment of vision-language features within the model. In addition, similar to real CXR reports, LLM-CXR often proposes valid causes for certain findings (e.g., suggesting aspiration as the cause of consolidation), demonstrating language-based reasoning ability characteristic of LLMs.[†]

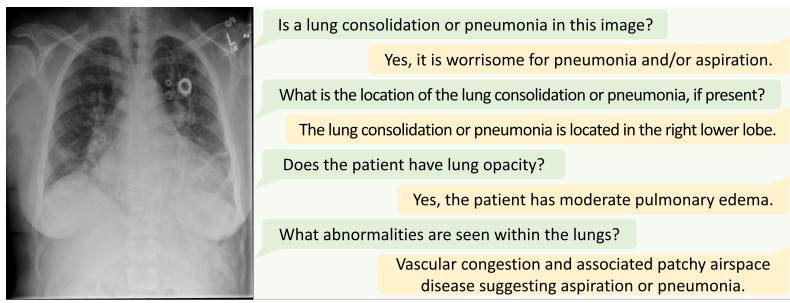

Figure 4: Examples of VQA with LLM-CXR. LLM-CXR understands questions given in natural language and is able to answer with relevant findings.[†]

To quantify the similarity between generated reports and ground-truth reports, we measured AUROC/F1 (Table 1) and Jaccard similarity index (Table 2) between labels of clinical significance (e.g. pneumonia, cardiomegaly) extracted from both generated and ground-truth reports using the CheXpert labeler (Irvin et al., 2019). Note that LLM-CXR and XrayGPT utilize input images with a resolution of 256×256 and 224×224 pixels respectively. In contrast, the original UniXGen model (UniXGen-512) uses images at a higher resolution of 512×512 pixels. Thus for comparison, we conduct experiments for UniXGen twice - once with its native 512px resolution and once again with 256px resolution images (used for LLM-CXR) upsampled to 512px.

In terms of both metrics, our model demonstrates superior performance when compared to competitors operating at the same resolution. Notably, there is no substantial performance gap observed when compared to UniXGen and RadFM, which operate with images at a higher resolution of 512×512 pixels. LLM-CXR (3B parameters) exhibits stronger accuracy than XrayGPT (based on the Vicuna-7B model). For completeness, we also measure the natural language generation performance metrics: BLEU, ROUGE-L, METEOR, CIDEr; however, because these metrics are less informative in the setting of medical image understanding where precise language is employed, we present them in the Appendix D (Table 5).

Table 1: CXR-to-report generation AUROC and F1.[1]

| AUROC ↑ | Atel. | Cnsl. | Pmtx. | Edema | Eff. | Pna. | Cmgl. | Les. | Frac. | Opac. | ECm. | NoF. | P.O. | Dev. | Micro | Macro | Weighted |
|---|---|---|---|---|---|---|---|---|---|---|---|---|---|---|---|---|---|
| RadFM | 0.587 | 0.498 | 0.503 | 0.633 | 0.657 | 0.504 | 0.611 | 0.516 | 0.498 | 0.514 | 0.502 | 0.666 | 0.499 | 0.597 | 0.638 | 0.556 | 0.596 |
| UniXGen-512 | 0.570 | 0.533 | 0.519 | 0.615 | 0.682 | 0.526 | 0.645 | 0.501 | 0.498 | 0.555 | 0.510 | 0.676 | 0.498 | 0.740 | 0.668 | 0.576 | 0.628 |
| IFCC | 0.479 | 0.508 | 0.486 | 0.504 | 0.496 | 0.486 | 0.545 | **0.518** | 0.498 | 0.497 | 0.463 | 0.497 | 0.499 | 0.494 | 0.543 | 0.497 | 0.498 |
| R2Gen | 0.501 | 0.485 | 0.504 | 0.500 | 0.503 | 0.502 | 0.505 | 0.510 | **0.500** | 0.501 | 0.511 | 0.494 | 0.500 | 0.498 | 0.542 | 0.501 | 0.500 |
| UniXGen-256 | 0.518 | 0.511 | **0.530** | 0.542 | 0.533 | 0.510 | 0.524 | 0.513 | 0.499 | 0.519 | 0.511 | 0.564 | **0.527** | 0.593 | 0.575 | 0.528 | 0.540 |
| XrayGPT | 0.551 | 0.506 | 0.511 | 0.590 | 0.595 | **0.519** | 0.570 | 0.511 | 0.499 | **0.553** | **0.539** | 0.592 | 0.490 | **0.646** | 0.617 | 0.548 | 0.577 |
| **LLM-CXR** | **0.558** | **0.517** | 0.496 | **0.619** | **0.641** | 0.509 | **0.577** | 0.506 | 0.494 | 0.537 | 0.505 | **0.677** | 0.498 | 0.640 | **0.628** | **0.555** | **0.597** |

| F1 ↑ | Atel. | Cnsl. | Pmtx. | Edema | Eff. | Pna. | Cmgl. | Les. | Frac. | Opac. | ECm. | NoF. | P.O. | Dev. | Micro | Macro | Weighted |
|---|---|---|---|---|---|---|---|---|---|---|---|---|---|---|---|---|---|
| RadFM | 0.325 | 0.024 | 0.018 | 0.404 | 0.494 | 0.034 | 0.387 | 0.065 | 0.000 | 0.177 | 0.026 | 0.524 | 0.000 | 0.381 | 0.370 | 0.204 | 0.341 |
| UniXGen-512 | 0.298 | 0.116 | 0.064 | 0.374 | 0.530 | 0.121 | 0.423 | 0.014 | 0.000 | 0.317 | 0.049 | 0.532 | 0.000 | 0.586 | 0.413 | 0.245 | 0.398 |
| IFCC | 0.159 | **0.083** | 0.020 | 0.203 | 0.312 | 0.006 | 0.270 | **0.068** | 0.000 | 0.323 | 0.042 | 0.200 | 0.000 | 0.292 | 0.220 | 0.141 | 0.225 |
| R2Gen | 0.168 | 0.018 | 0.020 | 0.073 | 0.129 | 0.034 | 0.263 | 0.043 | 0.000 | 0.240 | 0.051 | 0.289 | 0.000 | 0.254 | 0.201 | 0.113 | 0.183 |
| UniXGen-256 | 0.146 | 0.072 | **0.083** | 0.226 | 0.215 | 0.072 | 0.176 | 0.055 | 0.000 | 0.282 | 0.047 | 0.411 | **0.092** | 0.367 | 0.262 | 0.160 | 0.243 |
| XrayGPT | **0.279** | 0.065 | 0.049 | 0.334 | 0.404 | **0.110** | **0.347** | 0.058 | **0.016** | 0.352 | **0.076** | 0.371 | 0.000 | **0.470** | 0.326 | 0.209 | 0.330 |
| **LLM-CXR** | 0.272 | 0.081 | 0.013 | **0.382** | **0.464** | 0.084 | 0.327 | 0.036 | 0.000 | 0.278 | 0.035 | **0.535** | 0.000 | 0.453 | **0.360** | **0.211** | **0.350** |

Table 2: CXR-to-Report generation Jaccard similarity index.[1]

| JSI ↑ | Micro | Macro | Weighted | No mention | Possible | Negative | Positive |
|---|---|---|---|---|---|---|---|
| RadFM | 0.4894 | 0.2367 | 0.5568 | 0.6525 | 0.0123 | 0.0552 | 0.2270 |
| UniXGen-512 | 0.4716 | 0.2520 | 0.5464 | 0.6323 | 0.0423 | 0.0733 | 0.2602 |
| IFCC | 0.4141 | 0.1935 | 0.4877 | 0.5835 | 0.0027 | 0.0643 | 0.1237 |
| R2Gen | 0.4038 | 0.1988 | 0.4842 | 0.5797 | 0.0337 | **0.0701** | 0.1116 |
| UniXGen-256 | 0.5993 | 0.2438 | 0.6236 | 0.7480 | 0.0317 | 0.0449 | 0.1505 |
| XrayGPT | 0.4107 | 0.2206 | 0.4934 | 0.5773 | 0.0413 | 0.0690 | 0.1948 |
| **LLM-CXR** | **0.6092** | **0.2699** | **0.6420** | **0.7585** | **0.0483** | 0.0530 | **0.2198** |

[1] **Bold** in report-to-CXR task result tables indicates the best performance among models at the same resolution with LLM-CXR (256×256). If the highest performance is achieved at the 512, this is separately underlined.

## 3.2 CXR-VQA TASK

We adopt the VQA performance assessment framework of ELIXR (Xu et al., 2023b) which, in summary, asks about the presence, location, and severity of certain lesions or findings for each CXR image and marks each answer as 0 (the answer is incorrect, internally inconsistent, or irrelevant), 1 (correct), or 0.5 (partially correct or not quite correct but a reasonable explanation for the CXR image). First, as done in ELIXR, we randomly select eight cases that are labeled in MIMIC-CXR with the following diagnoses: 'No Finding', 'Pneumothorax', 'Pleural Effusion', 'Edema', 'Consolidation' OR 'Pneumonia' (considered a single unified class), and 'Lung Lesion'. We use the same questions and grading rubric used in Xu et al. (2023b). Note that the reported scores for ELIXR are taken from the paper Xu et al. (2023b) as their model is not publicly available, but the scores for XrayGPT and our model (LLM-CXR) were measured by the authors using open-sourced checkpoints.

Table 3: Accuarcy of the CXR-VQA task by topic and label diagnosis. ELIXR (Xu et al., 2023a) does not report its VQA accuracy by label diagnosis.

| Accuracy ↑ | All | Presence | Location | Size, severity, type |
|---|---|---|---|---|
| ELIXR | **54.8%** | **64.5%** | 41.0% | 25.0% |
| XrayGPT | 25.2% | 27.4% | 21.9% | 20.3% |
| RadFM | 32.7% | 34.5% | 31.3% | 20.8% |
| **LLM-CXR** | 44.8% | 41.3% | **50.0%** | **62.5%** |

| Accuracy ↑ | All | Cnsl./Pna. | Edema | Lsn. | NoF. | Eff. | Pmtx. |
|---|---|---|---|---|---|---|---|
| XrayGPT | 25.2% | 25.0% | 26.25% | 17.2% | 42.5% | 20.0% | 18.8% |
| RadFM | 32.7% | 34.4% | 31.3% | 40.6% | 61.3% | 26.3% | **23.8%** |
| **LLM-CXR** | **44.8%** | **39.1%** | **53.8%** | **50.0%** | **71.3%** | **53.8%** | 22.5% |

An example of VQA performed by LLM-CXR is shown in Figure 4. Accuracies of VQA in comparison with other multimodal LLMs capable of CXR reading are shown in Table 3. ELIXR (Xu et al., 2023a) uses PaLM-2 as the base LLM and uses the framework of BLIP-2 (Li et al., 2023) (*i.e.*, Q-former) for achieving vision-language alignment while XrayGPT Thawkar et al. (2023) uses a Vicuna-7B as the base LLM and uses the MedCLIP image encoder Wang et al. (2022c) and a linear mapping layer for vision-language alignment. Because the VQA task requires a general understanding of language, this task can only be done by the larger, LLM-based models. LLM-CXR holds promise even amongst bigger models.

## 3.3 REPORT-TO-CXR GENERATION TASK

Because our instruction-tuning includes image generation tasks, LLM-CXR is also able to generate matching chest X-rays when given a text report (Figure 5). We measure the quality of generated images with FID (Table 6 in Appendix D), and we measure vision-language alignment (*i.e.*, how well the text used to guide image generation is reflected in the generated image) by calculating the AUROC/F1 (Table 4) against the original CXR images in MIMIC-CXR-JPGs with a pretrained CXR lesion classifier network (Cohen et al., 2022), specifically, `densenet121-res224-all`. We compare our generated CXRs to RoentGen (Chambon et al., 2022), a stable diffusion-based model that generates CXRs based on text descriptions, and UniXGen (Lee et al., 2023), a bespoke non-LLM transformer-based model trained from scratch to generate CXR images and reports.

As shown in Figure 5, LLM-CXR is able to reflect lesion characteristics, location, and severity in its generated CXR images. Quantitatively, FID indicates that LLM-CXR generates images closer to real CXR images than UniXGen or RoentGen. With regards to alignment with input text in the generated images, AUROC/F1 indicates that LLM-CXR generates images that are most aligned with input texts.

Table 4: CXR generation AUROC and F1.

| AUROC ↑ | Atel. | Cnsl. | Pmtx. | Edema | Eff. | Pna. | Cmgl. | Les. | Frac. | Opac. | ECm. | Micro | Macro | Weighted |
|---|---|---|---|---|---|---|---|---|---|---|---|---|---|---|
| RoentGen | 0.7661 | 0.7535 | 0.6078 | 0.7084 | **0.8169** | 0.6054 | 0.7780 | 0.6283 | 0.6047 | 0.7162 | 0.7294 | 0.7061 | 0.7013 | 0.7055 |
| UniXGen | 0.7982 | 0.7509 | 0.6640 | 0.7876 | 0.7725 | 0.7065 | 0.7610 | 0.7200 | 0.7121 | 0.7867 | 0.7893 | 0.7435 | 0.7499 | 0.7518 |
| **LLM-CXR** | **0.8054** | **0.8263** | **0.7540** | **0.8111** | 0.8155 | **0.7722** | **0.7846** | **0.7852** | **0.7596** | **0.8311** | **0.8335** | **0.7907** | **0.7980** | **0.7991** |

| F1 ↑ | Atel. | Cnsl. | Pmtx. | Edema | Eff. | Pna. | Cmgl. | Les. | Frac. | Opac. | ECm. | Micro | Macro | Weighted |
|---|---|---|---|---|---|---|---|---|---|---|---|---|---|---|
| RoentGen | 0.8113 | 0.7286 | **0.7110** | 0.2954 | 0.7619 | 0.2501 | 0.7639 | 0.2677 | 0.6580 | 0.7781 | 0.7066 | 0.6578 | 0.6121 | 0.6298 |
| UniXGen | 0.8648 | 0.6903 | 0.4981 | 0.7378 | 0.7008 | 0.7213 | 0.7598 | 0.5606 | 0.6424 | 0.7794 | 0.7958 | 0.7164 | 0.7046 | 0.7082 |
| **LLM-CXR** | **0.8777** | **0.8283** | 0.7024 | **0.8061** | **0.8183** | **0.7529** | **0.8372** | **0.7678** | **0.7753** | **0.8342** | **0.8274** | **0.8065** | **0.8025** | **0.8054** |

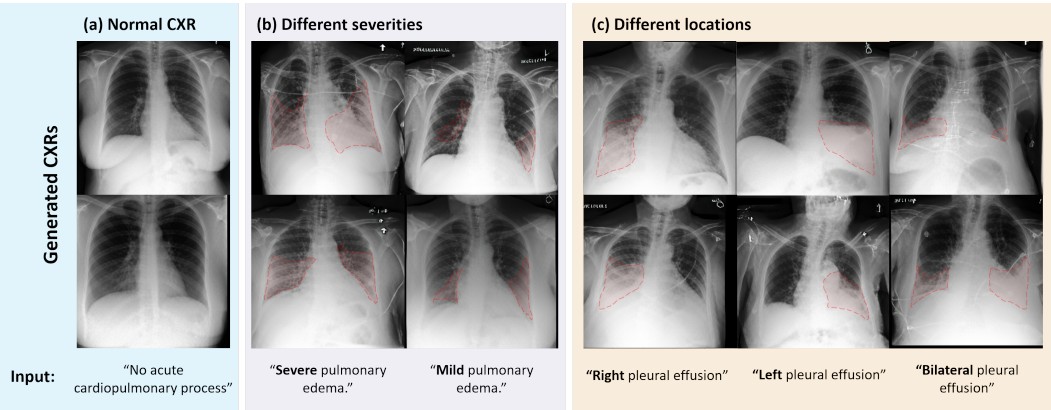

Figure 5: CXR images generated with LLM-CXR using radiology reports as input. **(a)** Normal CXRs. **(b)** Words such as "severe" and "mild" allow for the generation of different severities of lesions. **(c)** Specification of the location of lesions using words such as 'left', 'right', and 'bilateral'.

## 4 CONCLUSION

Multimodal LLMs have great potential to assist in the field of diagnostic radiology as they can reason about visual information and express their findings in natural language or images understandable by medical professionals. However, a major challenge is achieving sufficient vision-language alignment in these pretrained LLMs. Most work on vision-language alignment in LLMs has focused on developing adapter networks to connect an image-processing network and a pretrained, frozen LLM. However, this approach has so far fallen short in achieving the level of vision-language alignment needed to accurately describe medical images and is prone to hallucinations despite the use of LLMs with relatively high numbers of parameters. In this work, we proposed a different approach, an instruction-finetuning method for LLMs that enables them to understand and generate visual information, that shows more promise in achieving better-aligned vision and language features. We leveraged the language-understanding capability of LLMs to provide a complex training environment that induces the incorporation of visual features into its language features and shows that it can lead to better visual understanding and generation with LLMs even with a much smaller model.

## 5 LIMITATIONS AND FUTURE WORK

There are a few limitations of this study that could be improved upon. Most importantly, while our method shows better performance than other larger models, there is still much room to be improved in the alignment of visual and language features. For example, generated CXR reports still contain false positives (*i.e.*, they mention findings that are not actually present) and miss diagnoses. This problem could be further mitigated in the future by strengthening the alignment of images and text reports within the model by employing other vision-language techniques, improving the quality/quantity of the training data, or using larger LLMs. For instance, the radiology reports of the MIMIC dataset often refer to previous imaging studies which were unhelpful and act as noise instead of signal in our framework; we anticipate that using each patient's CXR scans longitudinally, *i.e.*, using the timestamps of each study, can help improve the quality of generated results by properly incorporating this information into the training process.

Furthermore, in our framework, a 256×256 image is translated into 256 image tokens. As a consequence, the resulting sequence contains relatively long token sequences compared to text, resulting in latency ranging from 30 to 60 seconds for image generation tasks and about 10 seconds for text generation tasks with the consumer GPU. Although our model presumably has faster inference times than larger models, it still cannot claim real-time responsiveness. A potential avenue for improvement is adopting techniques that enable dynamic tokenization of images (Jin et al., 2023), as opposed to using fixed-length tokens. This approach could potentially alleviate the latency issues and pave the way for more responsive real-time applications.

## ETHICS

Two important ethical issues at the intersection between medicine and AI are safety and privacy.

AI models will probably play increasingly larger roles in our healthcare systems. It is important that they are adopted in a way that improves patient safety and reliability of the systems already in place. LLMs have known issues with hallucinations and biases, which may be propagated when put into use without proper supervision. While communicative models like this one will potentially serve as a crucial interface between AI systems and human medical professionals so that such problems can be avoided, systems still need to be put in place so as to keep potential biases and hallucinations in check. There will also need to be continuous work to improve and scrutinize these models.

Furthermore, while our model is trained on the deidentified, publicly available MIMIC dataset, generative AI models such as ours raise concerns about privacy as institutions have the potential to develop these models with private patient data. When these multimodal LLMs reach the proficiency to be used in real clinics and become immediately valuable, regulations and technological security measures must be put in place to prevent breaches of patient privacy.

We hope that our model will serve as a step forward in developing reliable AI systems for healthcare.

## REPRODUCIBILITY

The pretrained models and datasets we use are all publicly available (Databricks' dolly-v2-3b, Imagenet-pretrained VQ-GAN, and MIMC-CXR-JPG). We release all code and model checkpoints upon publication along with step-by-step guidance to reproduce the methods explained in Section 2 so that anyone can reproduce our results[2].

## ACKNOWLEDGMENTS

This research was supported by the National Research Foundation of Korea(NRF)(RS-2023-00262527); Field-oriented Technology Development Project for Customs Administration funded by the Korean government (the Ministry of Science & ICT and the Korea Customs Service) through the National Research Foundation (NRF) of Korea under Grant NRF2021M3I1A1097910 & NRF2021M3I1A1097938; Korea Medical Device Development Fund grant funded by the Korea government (the Ministry of Science and ICT, the Ministry of Trade, Industry, and Energy, the Ministry of Health & Welfare, the Ministry of Food and Drug Safety) (Project Number: 1711137899, KMDF_PR_20200901_0015); Culture, Sports, and Tourism R&D Program through the Korea Creative Content Agency grant funded by the Ministry of Culture, Sports and Tourism in 2023; and Institute of Information & communications Technology Planning & Evaluation (IITP) grant funded by the Korea government(MSIT, Ministry of Science and ICT) (No. 2022-0-00984, Development of Artificial Intelligence Technology for Personalized Plug-and-Play Explanation and Verification of Explanation).

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

SUPPLEMENTARY MATERIAL

## A   IMPLEMENTATIONS DETAILS

**Dataset**   We used MIMIC-CXR v2.0.0 (Johnson et al., 2019a) as our dataset of CXR-report pairs. The data set consists of 377,110 CXRs from 227,835 radiology studies. The train-test split used the standard split of MIMIC-CXR-JPG (Johnson et al., 2019b). The test set sizes before and after pruning are 368,960 and 70,403, respectively. The test set was created using only AP/PA views from the raw data set excluding the train set, with a total of 3,530. The original images are files of various sizes, but the images were converted into JPEGs of square 256×256 images.

**Training VQ-GAN**   We trained the VQ-GAN (Esser et al., 2021) on 256×256 MIMIC-CXR train data starting from the pretrained weight and configuration of the `imagenet_f16_1024` model. For clinical information-preserving loss, 1024-dimensional features were extracted with TorchXRayVision's (Cohen et al., 2022) `densenet121-res224-all` model of the target image and the reconstructed image, and then the L2 distance of the two features was used. At this time, the loss was multiplied by 100 as the weight and added to the total loss. The number of indices in the codebook $K_{img}$ is 1024 and the dimension of the codebook embedding $n_z$ is 256. Since a 256×256 image is encoded and quantized with a 16×16 matrix by encoder and quantizer, the dimension of the quantized latent vector (*i.e.*, image tokens) of the image $d_z$ is 256 by flattening it. Model training was performed for 590k steps with the Adam (Kingma & Ba, 2014) optimizer, with a batch size of 2 and a learning rate of 4.5e-6.

**Fine-tunning LLM**   We used the `dolly-v2-3b` (Databricks, 2023) model, which is fine-tuned for the instruction-following task based on the GPT-NeoX (Black et al., 2022) architecture, as a base model. The model has a total of 2.8 billion parameters and has 50821 token types ($K_{text}$). We have extended the number of entries in the token embedding table to 51845 ($K_{text} + K_{img}$), as an additional 1024 ($K_{img}$) new image tokens should be available. For each image token from the VQ-GAN encoder, the value obtained by adding $K_{text}$ to each image token value is input to the LLM as a token ID. If the token output from the model is treated as an image token if the ID is greater than or equal to $K_{text}$, and each $K_{text}$ is subtracted and input to the VQ-GAN decoder.

Model training was performed with a learning rate of 5e-6 and a batch size of 16 using the AdamW (Loshchilov & Hutter, 2017) optimizer at all stages. Stage 1 uses 76k steps as 2 epochs and stage 2 uses 9k steps as 1 epoch for training. Training took about 14.5 hours for stage 1 and about 1.5 hours for stage 2 using NVIDIA A100 40GB ×8. The proportions of CXR-to-report, report-to-CXR, CXR-VQA, and NL-IF in the data set are [30%, 30%, 20%, 20%] and [21%, 21%, 63%, 5%], for stage 1 and stage 2, respectively.

## B   SYNTHETIC VQA GENERATION

We locally used LlaMa 2 (`Llama2-13b-chat-hf`) Touvron et al. (2023) to generate the synthetic VQAs. Radiology reports used were filtered from the MIMIC-CXR dataset through the method mentioned in the text, and a total of 27,322 were used. The following prompt was used to generate about 5 questions and answers using a chest X-ray report in MIMIC-CXR.

```
You are a radiologist asking questions about a chest X-ray image.
You always give long, detailed explanations as answers.

You are given a chest X-ray, delimited by triple backticks.
Create some questions
     about the chest X-ray, each question followed by an appropriate answer.

Output your questions and answers in JSON format like:
[{{"question": "<question1>", "answer": "<answer1>"}},
{{"question": "<question2>", "answer": "<answer2>"}}, ...].

```
{report}
```
```

```
Note, questions are never
     about any change from the last or previous chest X-ray scans or CTs.
Questions are also never
     about future plans; questions always focus on the chest X-ray itself.
Answers are very detailed and
     include explanations without repeating words that are in the question.

Here are the questions and answers:
```

We generated a total of 126,795 question-answer pairs using the described method. All question-answer pairs from stage 1, along with 75,100 pairs randomly extracted in stage 2, were employed for training purposes. The train-test split followed the MIMIC-CXR official split to which the source report belongs. Following are some examples of generated VQAs:

```
Q: What
     is the most likely cause of the subtle opacity at the right lung base?
A: Early pneumonia is the most likely cause of the subtle opacity
     at the right lung base, as there is no pleural effusion or pneumothorax.

Q: Is there any pleural effusion?
A: No, there is no pleural effusion present in the chest X-ray.

Q: Is there any pneumothorax?
A: No, there is no pneumothorax present in the chest X-ray.

Q: What is the appearance of the mediastinal silhouette and hila?
A: The mediastinal silhouette and hila appear normal in the chest X-ray.

Q: What is the impression based on the chest X-ray?
A: The impression based on the chest X-ray is subtle
     opacity at the right lung base, which could represent early pneumonia.
```

## C  TEMPLATE FOR LLM INSTRUCTIONS

Following is the template of the prompt for instruction fine-tuning from the Alpaca family which consists of *Instruction*, *Input*, *Response* sections. This template is employed consistently in both the instruction-following tuning process and the inference process. During inference, the model functions to predict tokens following the response key (### Response:).

```
Below is an instruction that describes a
     task. Write a response that appropriately completes the request.

### Instruction:
{instruction}
Input:
{input}

### Response:
{response}

### End
```

## D  ADDITIONAL EXPERIMENT RESULTS

Additional experimental results that were not covered in the main text due to space constraints are shown in Table 5 and Table 6.

Table 5: CXR-to-Report NLG Metrics.

| ↑ | BLEU1 | BLEU2 | BLEU3 | BLEU4 | METEOR | ROUGE_L | CIDEr |
|---|---|---|---|---|---|---|---|
| RadFM | 0.1344 | 0.0650 | 0.0362 | 0.0221 | 0.0867 | 0.1064 | 0.0264 |
| UniXGen-512 | 0.1471 | 0.0736 | 0.0403 | 0.0233 | 0.0813 | 0.1331 | 0.1413 |
| IFCC | 0.0636 | 0.0268 | 0.0106 | 0.0040 | 0.0809 | 0.0673 | 0.0058 |
| R2Gen | 0.0898 | 0.0359 | 0.0134 | 0.0057 | 0.0822 | 0.0734 | 0.0104 |
| UniXGen-256 | **0.1287** | **0.0570** | **0.0264** | 0.0137 | 0.0725 | 0.0893 | 0.0399 |
| XrayGPT | 0.1002 | 0.0417 | 0.0178 | 0.0075 | **0.1038** | 0.0907 | 0.0139 |
| LLM-CXR | 0.0920 | 0.0459 | 0.0260 | **0.0154** | 0.0690 | **0.1618** | **0.5248** |

Table 6: FID of generated CXRs from reports.[3]

| FID | inception-v3-2048 ↓ | txv-all-1024 ↓ |
|---|---|---|
| UniXGen | 78.19 | 7.894 |
| RoentGen | 42.38 | 6.039 |
| **LLM-CXR** | **22.75** | **0.7136** |

# E    ABLATION STUDIES AND DISCUSSION

We conducted a comprehensive ablation study to provide a rigorous justification for the design choices made in our method. This ablation analysis specifically focused on the CXR-to-report and report-to-CXR tasks and was evaluated using the same evaluation metrics as those outlined in the main text.

In this ablation study, we systematically removed one element at a time from our method to assess its impact. The factors subjected to ablation included the clinical information-preserving loss (CIP loss), simultaneous training of the CXR-VQA task (CXR-VQA), the use of the entire dataset by the additional training (stage 1 tr.) instead of using only pruned dataset, and the adoption of instruction tuning loss $L_{instruct}$ during fine-tuning as opposed to the joint loss $L_{joint}$ (instruct tr.).

Table 7: CXR-to-report generation AUROC and F1.

| | AUROC ↑ | | | F1 ↑ | | |
|---|---|---|---|---|---|---|
| | Micro | Macro | Weighted | Micro | Macro | Weighted |
| **LLM-CXR** | **0.6285** | **0.5553** | **0.5969** | **0.3604** | **0.2113** | **0.3504** |
| — CIP loss | 0.6170 | 0.5495 | 0.5853 | 0.3418 | 0.1994 | 0.3260 |
| — CXR-VQA | 0.6143 | 0.5492 | 0.5846 | 0.3378 | 0.1945 | 0.3173 |
| — stage1 tr. | 0.5770 | 0.5242 | 0.5435 | 0.2659 | 0.1347 | 0.2253 |
| — instruct tr. | 0.6071 | 0.5422 | 0.5745 | 0.3203 | 0.1927 | 0.3161 |

The findings from our CXR-to-report ablation study (Table 7) highlight the positive contributions of all ablation factors toward enhancing the model's performance. Notably, the incorporation of stage 2 training, which involves initially training with the full dataset, and the generation of the CXR-VQA dataset through augmentation, alongside simultaneous CXR-VQA task training, led to a substantial improvement in the alignment between images and reports within the CXR-to-report task.

This improvement can be attributed to several factors. Firstly, the significantly increased volume of image-report pairs during stage 1 training enabled more accurate learning of common image-report relationships, even though the additional dataset may not be directly related to our final tasks. Secondly, the CXR-VQA task provided direct supervision in comprehending and answering specific image characteristics, in contrast to the report generation task, where information about a single image is

---

[3] Txv-all-1024 is measured from the 1024-dim features of densenet121-res224-all from Cohen et al. (2022). Because we inevitably used our specific composition of MIMIC-CXR evaluation datasets for a common evaluation across multiple models and tasks, the results may differ from those reported.

distributed and represented. Consequently, these results suggest that the capacity to understand and respond to images acquired through the VQA task not only enhanced performance within the VQA task but also improved the overall quality of report generation.

Table 8: Report-to-CXR generation AUROC and F1.

| | AUROC ↑ | | | F1 ↑ | | |
|---|---|---|---|---|---|---|
| | Micro | Macro | Weighted | Micro | Macro | Weighted |
| **LLM-CXR** | 0.8065 | 0.8025 | 0.8054 | **0.7907** | **0.7980** | **0.7991** |
| — CIP loss | **0.8227** | **0.8196** | **0.8223** | 0.7890 | 0.7971 | 0.7977 |
| — CXR-VQA | 0.7505 | 0.7433 | 0.7465 | 0.7841 | 0.7871 | 0.7876 |
| — stage1 tr. | 0.7213 | 0.7125 | 0.7167 | 0.7217 | 0.7224 | 0.7231 |
| — instruct tr. | 0.5839 | 0.5751 | 0.5808 | 0.6040 | 0.5959 | 0.5965 |

In the case of the report-to-CXR task (Table 8), most design choices led to unambiguous improvements. However, the addition of the clinical information-preserving (CIP) loss shows conflicting effects by decreasing AUROC whilst increasing the F1-score. This may be due to better performance for minority classes with smaller support (e.g. pneumonia, enlarged cardiomediastinum) but lower performance for the more common majority classes (e.g. normal, pleural effusion). Taking into account that the incorporation of the CIP loss resulted in a substantial performance boost in the CXR-to-report task, we view this as a favorable trade-off within the constraints of the limited capacity of a small model such as dolly-v2-3b.

Table 9: FID of generated CXRs from reports.[3]

| FID ↓ | inception-v3-2048 | txv-all-1024 |
|---|---|---|
| **LLM-CXR** | 22.75 | **0.714** |
| — CIP loss | **20.93** | 0.931 |
| — CXR-VQA | 32.55 | 1.539 |
| — stage1 tr. | 32.05 | 1.226 |
| — instruct tr. | 21.51 | 1.477 |

We also measure FID score for generated CXR images (Table 9). All design choices lead to a decrease (improvement) in the FID score when measured using a CXR-specific classifier network (txv-all-1024). Certain techniques increase (worsen) the FID score measured using the Inception-V3 network. This is most likely due to the fact that our framework generates images that closely match the distribution of real CXR images so that features extracted from a network trained only on natural images such as the Inception-V3 cannot distinguish the subtleties in the generated images. Therefore, FID scores measured using txv-all-1024 are more appropriate for image quality assessment, and thus all techniques employed in the final model (LLM-CXR) can be interpreted as increasing generated CXR image quality.

## F  INSTRUCTIONS FOR MULTIMODAL TASKS

For the diversity of instructions, in the process of training and inference, one instruction is randomly sampled and used from the list of 10 instructions below. The instructions were modulated to 10 using OpenAI's ChatGPT (OpenAI, 2022) from the basic instruction.

**CXR-to-Report task**

- Generate free-text radiology reports for the entered chest X-ray images.
- Use the entered chest X-ray images to create corresponding free-text radiology reports.
- Based on the entered chest X-ray images, produce free-text radiology reports.
- Create free-text radiology reports that correspond to the entered chest X-ray images.
- Utilize the entered chest X-ray images to generate corresponding free-text radiology reports.
- Generate free-text radiology reports in accordance with the entered chest X-ray images.

- Use the entered chest X-ray images to create accurate free-text radiology reports.
- Produce free-text radiology reports that match the entered chest X-ray images.
- Create free-text radiology reports that are consistent with the entered chest X-ray images.
- Utilize the entered chest X-ray images to generate comprehensive free-text radiology reports.

**Report-to-CXR task**

- Generate a chest X-ray image that corresponds to the entered free-text radiology reports for the chest X-ray image.
- Use the free-text radiology reports for the chest X-ray image to produce a corresponding chest X-ray image.
- Utilize the entered free-text radiology reports for the chest X-ray image to create a corresponding chest X-ray image.
- Create a chest X-ray image that matches the free-text radiology reports entered for the chest X-ray image.
- Produce a chest X-ray image that is consistent with the free-text radiology reports entered for the chest X-ray image.
- Based on the free-text radiology reports for the chest X-ray image, generate a corresponding chest X-ray image.
- Use the free-text radiology reports entered for the chest X-ray image to create a corresponding chest X-ray image.
- Generate a chest X-ray image that is in accordance with the free-text radiology reports for the chest X-ray image entered.
- Create a chest X-ray image that corresponds to the free-text radiology reports entered for the chest X-ray image.
- Utilize the entered free-text radiology reports for the chest X-ray image to produce a corresponding chest X-ray image.

## G    REPOR-TO-CXR: MORE EXAMPLES

**Semantic descriptions of pathologies**    Radiology reports describe the semantic features of pathologies as they appear on the CXR scan. The most common descriptions involve location and severity. Here we show that our model incorporates features described in radiology reports when generating corresponding CXR images (Figure 6).

**Artificial devices**    Artificial devices are frequently captured in CXR images and reports. They have semantic features that are different from physiologic or pathologic features. We show that our model has learned to generate the general appearance of these devices (Figure 7).

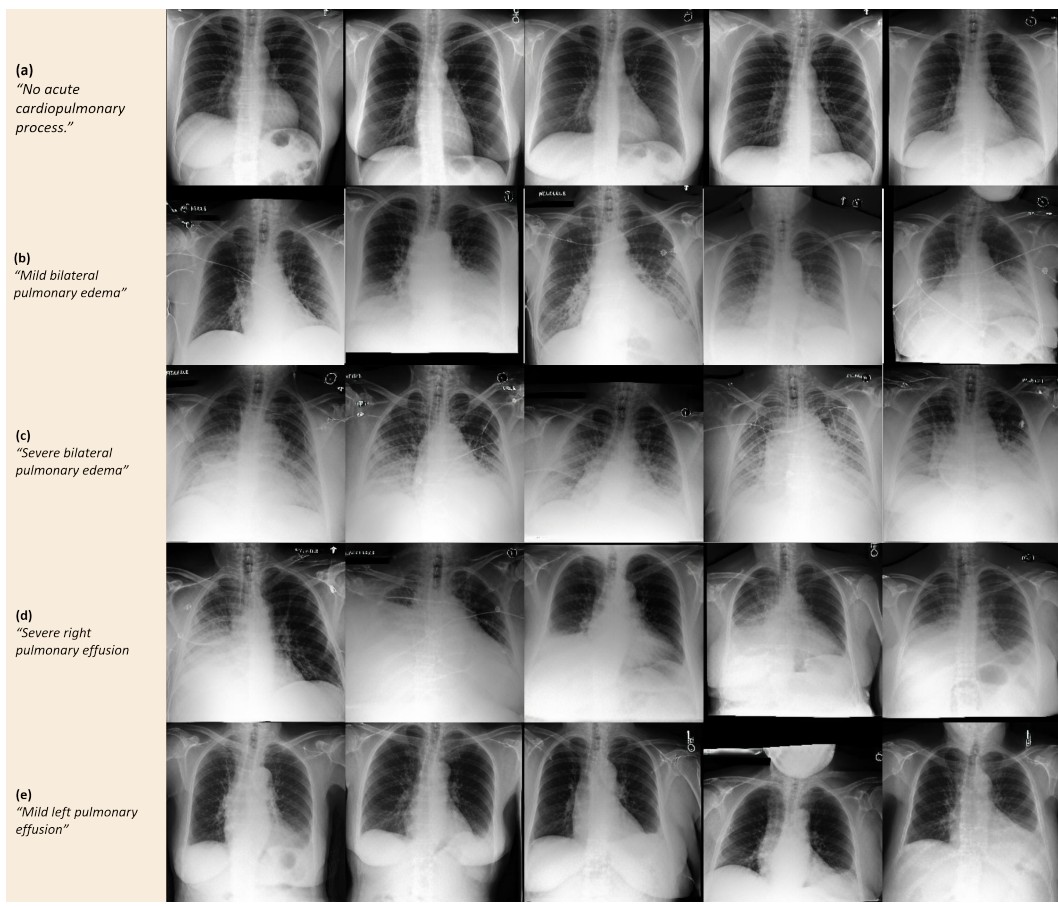

Figure 6: CXRs generated for different descriptions of pathologies. The model is able to accurately capture different levels of severity in the generated CXRs (b, c) and generate lesions in specified locations (d, e).

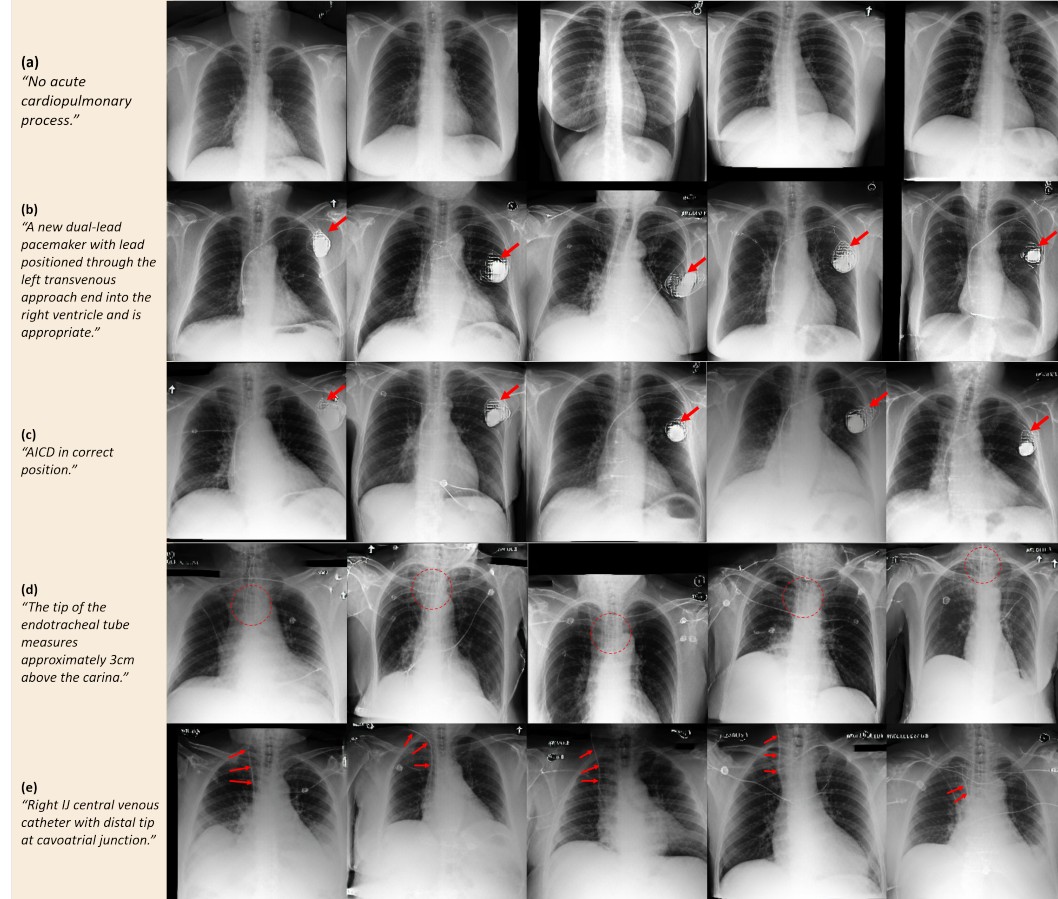

Figure 7: CXRs generated for radiology reports describing various foreign bodies. Reports of normal CXRs (a) and of large features such as pacemakers or AICDs (b, c) are realistically reflected in the generated images. Reports describing smaller, more detailed features such as endotracheal tubes and central venous catheters are represented in the generated CXRs but less accurately (d, e), with deterioration in image quality around the neighborhood of the described feature (*e.g.*, the trachea is not cleanly generated when the input report describes an endotracheal tube in (d)) or imperfect representation of feature itself (*e.g.*, venous catheters are generated but show missing parts in (e)).

## H   CXR-TO-REPORT: MORE EXAMPLES

| | Radiologist's Impression (GT) | Generated Impression (ours) |
|---|---|---|
| (a) | "...Low lung volumes. Areas of atelectasis at both left and right lung basis. No pleural effusions. No pneumonia" | "Low lung volumes with patchy and streaky opacities most likely reflective of atelectasis. Infection or aspiration cannot be excluded in the correct clinical setting..." |
| (b) | "Moderately severe pulmonary edema..." | "Moderate pulmonary edema." |
| (c) | "Mild pulmonary edema is stable ... Cardiomegaly is accentuated by the low lung volumes." | "Enlarged cardiac silhouette and interstitial pulmonary edema." |
| (d) | "Increased markings bilaterally may be due to the combination of underlying pulmonary fibrosis and moderate pulmonary edema, superimposed infectious process cannot be excluded." | "Slight increase in multifocal opacities which could be due to infection or mild pulmonary edema." |
| (e) | "Moderate cardiomegaly with AICD in unchanged position. No evidence of congestive heart failure or pneumonia." | "Pacer device leads terminating in the expected location of right atrium and right ventricle." |

Figure 8: Generated reports contain not only diagnoses but also descriptions of pathologies present in CXR images such as 'low lung volumes', and 'patchy and streaky opacities' (a). While the exact wording may differ from the ground-truth text report, generated reports are able to often capture the gist of the findings in the CXR images ("moderately severe" vs. "moderate" in (b); "cardiomegaly" vs. "enlarged cardiac silhouette" in (c)). Suggestions for potential pathologic processes that underlie the findings in the CXR also align with ground-truth reports (d). Generated reports also note the presence of artificial devices such as pacemakers (here, an AICD is recognized as a pacemaker as the distinction relies on finer details that would require further training to reliably distinguish (e)).[†]

