# OpenReview forum: "LLM-CXR: Instruction-Finetuned LLM for CXR Image Understanding and Generation"
_ICLR.cc/2024/Conference — ICLR 2024 poster_

### Official Review · Reviewer_MXmf · 2023-10-31

**Soundness:** 2 fair
**Presentation:** 2 fair
**Contribution:** 3 good
**Rating:** 5
**Confidence:** 4

**Summary:**

The authors proposed an image-text alignment framework for chest X-ray images and report pairs based on LLM models. In addition to the existing vision adapter, the authors reconstruct the adapted vision representation back to the images.  Furthermore, VQA pairs (as image-instruction-answer) are generated using GPT3.5 from the associated reports as a form of data augmentation for the image report pairs.  The reconstruction module is pre-trained as a VQGAN and then frozen when tuning the VQA instructions. The MIMIC-CXR dataset is employed here for the experiments. Superior results of the proposed model are reported in comparison to previous LLM models in report generation and VQA. However, the presented work suffers from several critical flaws that are detailed below.

**Strengths:**

+ Tackling the image-text alignment problem in medical imaging, which is not well-researched yet
+ The manuscript is overall easy-to-follow

**Weaknesses:**

- The authors claim the proposed bidirectional LLM is different from previous ones, as illustrated in Figure 1. However, I found the difference between them (a) and (c) is really minor. The encoder and decoder parts in the VQGAN are indeed equivalent to the vision adapter and image generative model, as listed in (a). Therefore, it is a bit overclaimed that it is novel to introduce bidirectional reconstruction tasks(both image-to-text and text-to-image) in the pre-training.
- The motivation for training an image-text aligned model is not clearly introduced and justified. First, it will be helpful to discuss how this model could be applied to the downstream clinical tasks. Then, the designed experiments only demonstrate the performance of the proposed method on these instruction tuning tasks, i.e., report generation based on images and VQA. It is hard to appreciate the benefit of adopting such a pre-trained and then SFTed model in practical use without demonstrating downstream applications. There are many chest X-ray benchmarks, and datasets are commonly used for the evaluation of pre-trained models, e.g., disease classifications and localizations.
- The metrics utilized in the experiments for most tasks are not the commonly used ones, e.g., AUCROC/F1 and Jaccard similarity index for report generation, FID alone for image generation, and accuracy alone for VQA.
- In the results of report generation, only LLM-based methods are compared. How about a dozen of those SOTA  methods in chest X-ray reporting? IU X-ray is another dataset commonly used for the evaluation of report generation performance. It will be helpful to report results on that as well, mainly when used as a cross-domain evaluation dataset.
- I am not sure why the upsampling is performed for the image data as described in section 3.1 since the X-ray images are much larger than this resolution. How this upsampling process will affect the results?
- In section 2.2, the authors mentioned that the image tokens are parts of expanded embedding tables. I wonder how big K_img should be?

**Questions:**

see weaknesses

---

> ### Author Response · Authors · 2023-11-20
> **Response to reviewer MXmf**
>
> Thank you for your thorough reading and insightful comments. We address your concerns below.
>
> **W1. It is overclaimed to introduce bidirectional reconstruction tasks in pre-training as a novel approach.**\
> We would like to more clearly delineate the differences between Figure 1’s (a) and (c). The image encoder and decoder in (a) have already been pre-trained with the paired vision-language dataset; and the encoder outputs pseudo-text, which the decoder in turn receives as input. In this process, the language model does not directly engage with visual information as it takes pseudo-text as input and outputs pseudo-text (this is why the LLM can remain frozen in methods like BLIP-2).
>
> In our approach, visual information is passed directly to the LLM without passing through a vision-language pretrained network. The encoder simply performs tokenization of the image. LLM thus receives direct visual information, and vision-language alignment occurs within its own parameters on top of its pretrained language features. We believe that, as shown by our results, this is a promising avenue of multimodal LLM research that deserves attention separate from the currently dominating method of using vision adapter networks to implement multimodal LLMs.
>
> **W2. Motivation not clear. Explanation of downstream tasks?**\
> Direction to AGI development will likely involve integration of a central LLM with various expert models operating as modules; and we envision that communication between them will heavily utilize natural language. From this perspective, expert models should have language-based communication skills. As for a specific downstream example task, models like LLM-CXR can be instructed to find patients with specific findings described by text. For example, if the inclusion criteria for a clinical trial calls for ‘patients with severe bilateral pleural effusions on CXR’, the model can be used as an agent to assist this process.
>
> **W3. Metrics used are uncommon.**\
> We respectfully disagree on this point. We use AUROC/F1-scores based on label extraction to assess the accuracy of report generation because it is most commonly used in this area of research as shown in [1, 2 3]. Jaccard similarity index was also employed because AUROC/F1-scores favor detection of positive classes whereas the Jaccard similarity index treats both positive and negative classes equally; this is important as hallucinations are a well-known problem in multimodal language models, and negative classifications are of equal importance to positive classifications in analyzing medical images. Although we find them much less informative for medical reports, we have also added NLG metrics (e.g. BLEU, ROUGE, CIDEr).
>
> For the image generation task, FID assesses quality of generated images, and AUROC/F1 using a pretrained CXR image classifier assesses whether generated images are in line with the given text inputs used to generate them. The latter evaluation is more important in our case as it assesses image-text alignment. To our best knowledge, there is no common or agreed upon way to evaluate the alignment between generated images from a model and the given input prompts used to generate them. We believe that this set of assessments, combined with the provided qualitative examples, showcases the capabilities of the models under investigation.
>
> **W4-1. Only LLM-based models are compared for report generation**\
> Med-PaLM1/2 would also be a great target for comparison, but they are not publicly available yet. As per your suggestion, we include radiology report generation models that have the best reported performance as baselines for comparison. namely IFCC [3] &  R2Gen [4]. These models showed weak performance compared to LLM-based models.
>
> ***Table 1. CXR-to-report (part)***
> |F1|**Micro**|**Macro**|**Weighted**|
> |-|-|-|-|
> |RadFM|0.370|0.204|0.341|
> |LLM-CXR|0.360|0.211|0.350|
> |IFCC|0.220|0.141|0.225|
> |R2Gen|0.201|0.113|0.183|
>
> **W4-2. IU X-ray dataset evaluation**\
> Please refer to general response.
>
> **W5. Upsampling in section 3.1**\
> We wanted to simulate a situation where UniXGen, which can only receive 512px images, receives 256px images (the resolution for input to LLM-CXR and XrayGPT). But we also report the outcomes for UniXGen using its native resolution 512x512 images in Table 1.
>
> **W6. How big should the image token space be?**\
> $K_img$ is a value that is predefined as a design configuration for the VQ-GAN model architecture. We used the $K_img=1024$ as  used in the original VQ-GAN paper for 256px images.
>
> [1] Lee et al. "UniXGen: A Unified Vision-Language Model for Multi-View Chest X-ray Generation and Report Generation". arXiv 2023.\
> [2] Li et al. "Dynamic Graph Enhanced Contrastive Learning for Chest X-ray Report Generation". CVPR 2023.\
> [3] Delbrouck et al. "Improving the Factual Correctness of Radiology Report Generation with Semantic Rewards". EMNLP 2022.\
> [4] Chen et al. "Generating Radiology reports via Memory-driven Transformer". EMNLP 2020.

---

> > ### Comment · Reviewer_MXmf · 2023-12-04
> > **Most of the concerns are well-addressed in the rebuttal**
> >
> > I thank the authors for the detailed reply and am sorry for the delay (did not receive emails like in previous years). Based on the explanation of the motivation and possible practical use and the added experimental results, I would like to raise my rating.

---

> ### Author Response · Authors · 2023-11-23
> **[Reminder] Summarization of our rebuttal**
>
> Dear reviewer MXmf,
>
> We believe that we have addressed the concerns you have raised. Specifically,
>
> 1. We **clarify the distinction from previous approaches** in that encoders in previous works are vision-language pretrained and output pseudo-text an LLM can already understand, whereas in our approach the encoder serves to simply tokenize the image using a VQGAN encoder trained only on images and output image features that the LLM itself must learn.
> 2. We explain our envisioned downstream use cases.
> 3. We **outline the detailed context of the employed metrics** used to evaluate models in the paper and explain with citations that they are commonly used by other similar works as well.
> 4. We provide **results to further experiments** you suggested (*i.e.,* comparison to non-LLM-based radiology report generation baselines and evaluation on the IU chest X-ray dataset).
> 5. We answer your questions about about image upsampling in section 3.1 and size of the image token space
>
> As the deadline for the reviewer-author discussion phase is fast approaching (there is only a day left), we would like to extend a warm reminder and ask whether we have addressed your questions and concerns adequately. We would be happy to clear up any additional questions.
>
> Best regards, Authors.

---

### Official Review · Reviewer_DPiA · 2023-10-31

**Soundness:** 3 good
**Presentation:** 4 excellent
**Contribution:** 3 good
**Rating:** 8
**Confidence:** 2

**Summary:**

This work developed an instruction-finetuning method to integrate visual information into out-of-the-box LLMs, which can be used for bidirectional multi-modal CXR tasks, such as CXR report generation, VQA, and report-to-CXR generation.

**Strengths:**

- The image is tokenized by VQ-GAN, ensuring that clinical information is preserved. It is an efficient use of existing resources and knowledge by expanding pre-trained LLM's embedding space to include image tokens.
- The bidirectional instruction fine-tuning maintains the integrity of the LLM's structure and objectives while expanding its capabilities, which have many applications in the field.

**Weaknesses:**

The interpretability and scalability are not discussed.

**Questions:**

- How scalable is the tokenization and fine-tuning process for larger medical images (e.g., 3D CT/MR scans)?
- How interpretable are the model's decisions, especially given the clinical context where explanations for predictions are crucial?
- The comparison with other related methods is missed. https://github.com/chaoyi-wu/RadFM

---

> ### Author Response · Authors · 2023-11-20
> **Response to reviewer DPiA**
>
> Thank you for your valuable review and positive feedback. We address your concerns below.
>
> **Q1. How scalable is the tokenization and fine-tuning process for larger medical images (e.g. 3D/MR scans)?**\
> Although we have not conducted experiments for 3D data, we presume that 3D scans will probably need a similar tokenization strategy adapted to be more appropriate for higher-dimensional data. We think that certain ideas from our approach can be extended to 3D images as well; for example, the TXV-loss we employed to preserve clinical information during tokenization would also be applicable for 3D medical images.
>
> **Q2. How interpretable are the model's decisions?**\
> We agree that interpretability is a crucial consideration in clinical models. We believe that the natural language-based outputs of LLM-CXR intrinsically offer a higher level of interpretability than traditional computer vision models that output classification results. LLM-CXR is also able to generate images given some input text like ‘severe left-sided pneumonia’; this offers a window into the inner workings of the model as we can ask it to show images that correspond to a certain concept described using text.
>
> As an additional note, compared to approaches in which the LLM receives images in the form of pseudo-text produced by an adapter network (e.g. image encoding networks), we believe that our method of directly processing images and treating them equal to natural language is more promising for interpretability as if the semantics of the image are delivered as pseudo-text, interpretation of the LLM’s understanding of images themselves is bound to be limited.
>
> **Q3. Comparison with RadFM**\
> Thank you for bringing RadFM model to our attention. This work shares many of our motivations, and we were excited to see that they have open-sourced their work. We include RadFM in our assessments for CXR-to-report generation and CXR-VQA:
>
> ***Table 1. CXR-to-report generation (part)***
> |**F1**$\uparrow$|**Micro**|**Macro**|**Weighted**|
> |---|---|---|---|
> |RadFM|**0.370**|0.204|0.341|
> |XrayGPT|0.326|0.209|0.330|
> |LLM-CXR|0.350|**0.211**|**0.350**|
>
> ***Table 3. CXR-VQA (part)***
>
> |**Accuracy**$\uparrow$|**All**|**Presence**|**Location**|**Size,severity,type**|
> |---|---|---|---|---|
> |XrayGPT|25.2%|27.4%|21.9%|20.3%|
> |RadFM|32.7%|34.5%|31.3%|20.8%|
> |**LLM-CXR**|**44.8%**|**41.3%**|**50.0%**|**62.5%**|
>
> Note that RadFM is based on the LLaMa-13B model and LLM-CXR is based on the dolly-3B model, so there is about a 4-fold difference in parameter count; but LLM-CXR's performance is comparable if not better.

---

> > ### Comment · Reviewer_DPiA · 2023-11-22
> >
> > Thanks for your feedback.
> >
> > > we presume that 3D scans will probably need a similar tokenization strategy...
> >
> > 3D scans usually have diverse sizes and spacings, which are important physical metadata for medical images. I don't think the proposed tokenization strategy is directly applicable. It would be great if the authors could add more discussions on this point in the manuscript.
> >
> > - interpretability
> >
> > I agree that natural language-based outputs are more intuitive than traditional classification models but I don't think this is the interpretability. Healthcare is an evidence-based field. Instead of directly generating the report,  what we expected is that the model could automatically detect all pathological findings, followed by classifying these findings into meaningful categories.
> >
> > - Comparison with RadFM
> >
> > It is impressive that the proposed method can obtain better performance with fewer parameters.
> >
> > Overall, I think the contribution is enough as a conference paper. I still suggest acceptance.

---

> ### Author Response · Authors · 2023-11-22
>
> > Overall, I think the contribution is enough as a conference paper. I still suggest acceptance.
>
> Thank you for the recognition of our contributions and for the optimistic suggestions based on domain expertise.
>
> After reading your review and the RadFM paper, we have also begun work on 3D applications of this methodology. Our past experience informs us that methods using 3D volumes directly is vulnerable to the 'curse of dimensionality' (data requirements increase exponentially with increase in dimensionality); and so we are considering a "2.5D" approach in which a CT or MRI volume is treated as a sequence of 2D slices, each of which can be tokenized using methodologies developed for 2D images. While results still remain to be seen, we will, as per your suggestion, add discussions on this topic in the manuscript as it may be of interest to more readers.
>
> As for interpretability, we are also developing methods to offer more insight into model outputs such as visualizing the parts of the image that are most influential in generating certain natural language phrases (mimicking how radiologists often mark the important parts of a scan with arrows or circles). Since LLM-CXR effectively translates information between images and text within a single neural network (the LLM) and not in disparate adapter modules, we believe that making this translation process explicit will be most feasible for our model particularly. As you mentioned, because evidence-based care and clear rationales for decision-making are crucial in healthcare, we see this as important follow-up work.
>
> Thank you again for your encouraging assessment. We were genuinely excited with this discussion. Please let us know if you have any further suggestions or ideas.

---

### Official Review · Reviewer_FRze · 2023-11-01

**Soundness:** 3 good
**Presentation:** 2 fair
**Contribution:** 2 fair
**Rating:** 6
**Confidence:** 4

**Summary:**

The paper delves into enhancing Large Language Models (LLMs) with vision-language capabilities, specifically targeting medical imaging like chest X-rays (CXR). Recognizing that current "adapter network" methods might limit the deep integration of visual and language features, the authors propose a novel "instruction-finetuning" method. Drawing from vision-language pretraining (VLP) techniques, they tokenize images using VQ-GAN, facilitating the generation of combined text and image sequences. Rather than building a new model, they finetune a pretrained LLM with diverse CXR-related instructions. This approach allows the LLM to understand and generate visual data without structural modifications. Their finetuned LLM showcases proficiency in tasks such as translating CXRs to reports and vice versa, and performing CXR-specific visual question answering. The model not only outperforms specialized models in these tasks but also demonstrates the potential of seamlessly integrating visual and language abilities in LLMs for medical applications.

**Strengths:**

Strengths:
1. **Novel Approach**: The paper introduces a novel "instruction-finetuning" method, which is a significant departure from the prevalent "adapter network" techniques. This innovative approach allows for a more intimate integration of visual and language features in LLMs.

2. **Leveraging Existing LLMs**: Instead of starting from scratch, the authors smartly utilize the inherent instruction-following abilities of pretrained LLMs. This approach is efficient and maximizes the potential of existing models.

3. **Broad Application**: The finetuned LLM exhibits versatility in handling a range of tasks, from converting CXRs to reports, generating CXRs from textual reports, to CXR-specific visual question answering. This breadth of application showcases the model's potential in real-world medical scenarios.

4. **Outperformance**: The paper demonstrates that their model surpasses other specialized models in various tasks. This comparative analysis underscores the efficacy of their approach.

5. **Seamless Integration**: By using VQ-GAN for image tokenization, the authors ensure a smooth integration of image and text token spaces without necessitating structural changes to the base LLM. This seamless integration is crucial for practical implementations.

**Weaknesses:**

Areas for improvement:

1. **Potential for Catastrophic Forgetting**: As with any endeavor to expand a pretrained model's capabilities, there's the inherent risk of the model losing or diminishing its foundational skills—known as catastrophic forgetting. While the paper's intent is to preserve and build upon the LLM's language capabilities, it doesn't extensively address measures taken to prevent this potential degradation.

2. **Adapter Network Dismissal**: The paper critiques the prevalent "adapter network" approach but doesn't provide a comprehensive empirical comparison. A deeper, side-by-side evaluation highlighting performance, adaptability, and computational efficiency would offer a clearer picture of why their "instruction-finetuning" method is superior or preferable.

3. **VQ-GAN Limitations**: Using VQ-GAN for image tokenization introduces its own set of challenges. VQ-GANs, while powerful, can sometimes produce artifacts or representations that aren't entirely faithful to the original image. The paper doesn't discuss how it mitigates or addresses these potential shortcomings, leaving room for questioning the quality or accuracy of the generated content.

4. **Lack of Diverse Evaluation**: The paper's evaluation, though rigorous, might benefit from a more diverse set of metrics. For instance, qualitative evaluations or user studies involving medical professionals could provide insights into the model's practical utility. Comparisons with human diagnostic capabilities might also offer a benchmark for the model's proficiency.

5. **RoentGen AUC and FID numbers**: The authors need to address the discrepancy between the AUC and FID numbers provided in their paper and those in the RoentGen paper. It should be clarified whether these numbers were reproduced by the authors or cited from the RoentGen study, and if there are differences, the paper should explain the reasons behind these.

6. **Minor edits**:

a. Grammar throughout the text requires attention for better readability, as exemplified by the construction of the second sentence in section 2.2.

b. The term "txv-all-1024" utilized in the text should be clearly defined. It is presumed to refer to the 1024-dimensional outputs from a densenet-121 model trained on chest X-ray classification, but this assumption needs confirmation in the paper for clarity.

**Questions:**

If the authors address the areas of improvement, I can reconsider my assessment.

---

> ### Author Response · Authors · 2023-11-20
> **Reply to reviewer FRze**
>
> **W1. Potential for catastrophic forgetting.**
>
> **A.** Our main goal was to leverage the pre-existing language capabilities of a large language model to tune it into an (instructionable in natural language) expert model that can describe images using text, generate images given an input text, and perform visual question-answering tasks. Because all three tasks and natural language instruction following ability require an understanding of language, we build on top of an LLM; as we simply sought to leverage the language capabilities of the LLM, it was not our goal to maintain all of the previous text-based knowledge contained in the model.
>
> Our model is intended to be a form of expert module for future AGI that communicates in natural language, and therefore we believe that natural language skills, excluding task-related language skills, only require a basic ability to follow natural language instructions.  While it would of course be beneficial to mitigate forgetting as much as possible in a model with a given number of parameters, we view this as future work that requires its own set of innovations.
>
> **W2. Lack of side-by-side comparison with models that use adapter networks that justifies the claim that our approach is better.**
>
> **A.** We do not mean to dismiss approaches using adapter networks for vision-language integration. Most concurrent vision-language research works use these adapter networks (e.g. XrayGPT and ELIXR) for integration of visual information to LLMs. On the other hand, there has been a surprising lack of investigation into other novel avenues (such as our method of utilizing instruction-finetuning for vision-language alignment). Moreover, the reviewer is kindly reminded that the adapter networks cannot generate images without connecting to an image generation model such as Stable Diffusion.
>
> In the medical domain especially, we believe that our method of instruction-tuning to achieve alignment of image-text information may hold more promise as pixel-level detailed visual information must be adequately described in the text. This is in contrast to natural images where, as shown in works such as BLIP-2, the use of adapter networks to pass information to an LLM may be enough to yield sufficiently performant models as broader features confer the semantic meaning. This was an important motivation behind our work, and we wanted to convey this need for investigation of alternative approaches to image-text alignment rather than dismissing the adapter network approaches as a whole.
>
> Our evaluation experiments comparing results for LLM-CXR to those of XrayGPT and ELIXR showcase the viability of our approach against these adapter network approaches, as they both utilize larger models but show similar or lower performance in image understanding.
>
> **W3. VQ-GAN may produce images that are not faithful to the original image.**
>
> **A.** As pointed out, VQ-GAN has limitations in reconstruction due to its characteristics (e.g. vector quantization). There are limitations in terms of information preservation, so VQ-GAN uses LPIPS-like loss to improve the preservation of perceptual information. An important informational aspect of CXRs is the presence or absence of certain lesions. Therefore, we introduced a clinical information-preserving loss into VQ-GAN to preserve lesion information as much as possible in the VQ-GAN structure.
>
> Considering that our ultimate goal is not to ensure that LLM accurately restores a given image but to understand images and generate them based on text, it was more in line with our objective to aim for latent tokens that contain semantic information rather than details for exact reconstruction. In other words, in our work, we used VQ-GAN's reconstruction loss as a means of encoding and decoding an image for textual understanding rather than for complete reconstruction.
>
> As a side note, we would like to point out that an adapter network with Stable Diffusion image generator cannot reconstruct original input images as closely as our VQ-based method.
>
> **W4. More diverse metrics e.g. human evaluations of quality and diagnostic performance evaluations against human radiologists.**
>
> **A.** The comprehensive evaluation by the end-users (ie, clinicians) would be the eventual target, but owing to time constraints, we cannot conduct these experiments yet. While we will try to conduct such experiments, we believe that the utilized metrics can guide further model development.
>
> **W5. Discrepancies in AUC/FID with that reported in the RoentGen paper.**
>
> **A.** The Difference is most likely because we used a different test set. To compare the different models from different works, we used a single test set (that is presumably different from the test set used in the RoentGen paper) across all experiments. However, we did use the model weights and inference code provided by the authors.
>
> **W6. Minor errors in writing.**
>
> **A.** We have fixed and improved the writing to be clearer to the reader.

---

> > ### Comment · Reviewer_FRze · 2023-11-22
> > **Thank you for addressing my concerns**
> >
> > Thank you authors for addressing my concerns. Based on the authors' reply, I will update my score to 6.

---

> > > ### Author Response · Authors · 2023-11-22
> > >
> > > > Thank you authors for addressing my concerns. Based on the authors' reply, I will update my score to 6.
> > >
> > > Thank you for your careful consideration of our work, and we appreciate your positive feedback. We are excited that the discussion process has yielded visible improvements to the paper. Your encouraging words motivate us to continue pursuing this direction of research. We are also committed to addressing any further suggestions you may have, so please let us know if you have any further comments.

---

### Official Review · Reviewer_xBeK · 2023-11-03

**Soundness:** 3 good
**Presentation:** 2 fair
**Contribution:** 2 fair
**Rating:** 6
**Confidence:** 4

**Summary:**

The manuscript introduces an instruction-tuning technique geared towards amplifying the image comprehension and generative capabilities of a text-exclusive LLM for CXR imagery interpretation and generation. The outcomes indicate a top-tier performance in both image-text understanding and generative capacities, surpassing earlier models in the field.

**Strengths:**

The method of instruction-finetuning effectively elevates the LLM's capability to intricately map vision-image modalities. Consequently, it markedly excels in tasks like CXR-to-Report generation, CXR-VQA, and Report-to-CXR generation, outshining the open-sourced versions of models such as XrayGPT and UniXGen.

**Weaknesses:**

1. The paper's premise relies on the fine-tuning of LLM across multiple instruction tuning tasks. This approach isn't particularly novel, as numerous large-scale medical models adopt a similar strategy but with a broader functional range. Both the innovation in method and the paper's applicative contribution seem to be lacking.
2. The LLM-CXR, when compared with the open-sourced versions of XrayGPT and UniXGen, appears to be unfair. It's noted that these models aren't trained on a consistent instruction tuning dataset. It's recommended that a uniform dataset is utilized for training before making such comparisons. Furthermore, there's an evident absence of comparison with existing LLM+instruction tuning models, such as Med-PaLM and Med-PaLM 2.
3. The paper's comparative methods across various understanding and generative tasks aren't comprehensive. For instance, in the realm of report generation, several contemporary methodologies exist, such as "Dynamic Graph Enhanced Contrastive Learning for Chest X-ray Report Generation, CVPR 2023."
4. The evaluation metrics used for report generation seem lacking in depth. For instance, within the paper (table 1), only the AUROC scores across six categories are reported, while prior research typically reports the average Precision/Recall/F1-score across all 14 categories.
5. The paper's writing style isn't fluid and contains multiple errors, hindering smooth reading and comprehension. For instance, repetitive phrases like "… generate these VQAs as shown as shown in …" and grammatical mistakes like "During the fine-tuning process …" detract from the paper's clarity.

**Questions:**

Methodology Concerns:
1. How does the instruction-finetuning method differentiate itself from existing large-scale medical models that employ a similar strategy?

Comparative Analysis:
1. Is the comparison between LLM-CXR and the open-sourced versions of XrayGPT and UniXGen made on a consistent dataset?
2. Why hasn't the paper compared its methodology with existing LLM+instruction tuning models, such as Med-PaLM and Med-PaLM 2?
3. Are there reasons for not exploring contemporary methodologies in report generation, like "Dynamic Graph Enhanced Contrastive Learning for Chest X-ray Report Generation, CVPR 2023"?

Evaluation Metrics:
Why were the AUROC scores for report generation only presented for six categories instead of the standard 14 categories?

**Details Of Ethics Concerns:**

The MIMIC Data Use Agreement explicitly prohibits sharing access to the MIMIC data with third parties, including sending it through APIs provided by companies like OpenAI, or using it in online platforms like ChatGPT. (https://physionet.org/news/post/415)

---

> ### Author Response · Authors · 2023-11-20
> **Reply to reviewer xBeK**
>
> First and foremost, we would like to thank you for catching an error in our data processing process (use of the OpenAI API with MIMIC-CXR data) before publication. We hope to address your other concerns below.
>
> **W1/MethodologyConcerns-Q1.  The approach of multiple instruction-tuning tasks is not particularly novel. How does our methodology differentiate itself from other existing large-scale models that employ a similar strategy?**
>
> **A.** The main distinction of our work from other work using instruction-tuning to build medical LLMs is that we use instruction-tuning to teach an LLM to both understand and generate images. Most instruction-tuning methodologies are geared towards changing a next word-predicting model to an instruction-following model using instruction-answer pairs (e.g. GPT3.5, PaLM) rather than promoting the integration of visual information to language-based models. There is previous work on instruction-tuning with visual information (e.g. LLaVA, InstructBLIP) which are most similar to our work; but these works employ adapter networks CLIP and BLIP respectively for image-text alignment and employ instruction-tuning techniques simply as a means of passing visual information to an LLM. We on the other hand aimed to use instruction-tuning itself as a method of aligning images and text. You may have noticed that no separate image-text aligning techniques (e.g. CLIP, BLIP, ALIGN) are used at all. In more detail, image encoders in approaches such as LLaVA and InstructBLIP have already been pre-trained with the paired vision-language dataset, and the encoder already outputs pseudo-text; the decoder in turn receives pseudo-text as input. In this process, the language model does not directly deal with images as it takes pseudo-text as input and outputs pseudo-text. However, in our approach visual information is passed directly to the LLM, and the LLM itself learns visual features and vision-language alignment on top of its previous knowledge of the language.
>
> Another distinction is that our instruction-tuning method also enables the generation of images. This confers a few benefits: 1) we can ask the model to directly visualize a certain concept encapsulated in text (e.g., pneumonia); 2) this bidirectionality in generation turns out to be vital for aligning images and text without any adapter networks; 3) the visual instruction tuning method of LLaVA can teach the model image features but only the features that are close to the semantic meaning of the texts and not the fine-grain visual information in images that are not completely described by text.
>
> We hope that the distinction of our work from previous ones is now clearer.
>
> **W2/ComparativeAnalysis-Q1. Comparison with XrayGPT and UniXGen is not fair because they are not trained on the same instruction-tuning dataset.**
>
> **A.** XrayGPT and UniXGen are trained on the same dataset as ours (MIMIC-CXR). However, we have used the dataset to train in an instruction-tuning manner.
>
> Regarding your comment on fair comparison, we respectfully disagree with the reviewer since we started our research from the viewpoint that using an enriched instruction-tuning set itself is a distinction from previous works such as XrayGPT and UniXGen.
>
> **W3/Comparative-AnalysisQ2/3. Evaluation is not comprehensive. Compare with contemporary radiology report generation methodologies or Med-PaLM1/2.**
>
> **A.** The motivation for our work was to teach a large language model to understand and generate chest X-ray images so that it can serve as a more general-purpose interface for medical images than report generation models. This is why we felt most meaningful comparisons were against similar large language models that can understand CXR images (XrayGPT, ELIXR).
>
> As per your suggestion, we have included radiology report generation models with best reported performance as additional baselines for comparison. The results confirm that our models are superior or comparable to the dedicated contemporary report generation models (Table 1, 2 in the revised manuscript).
>
> ***Table 1. CXR-to-report (part)***
> |F1|**Micro**|**Macro**|**Weighted**|
> |-|-|-|-|
> |RadFM|0.370|0.204|0.341|
> |LLM-CXR|0.360|0.211|0.350|
> |IFCC|0.220|0.141|0.225|
> |R2Gen|0.201|0.113|0.183|
>
> As you mentioned, Med-PaLM1/2 would also be a great target for comparison, but they are not publicly available yet. Also, Mingjie Li *et al.* did not release pretrained checkpoints, and the training code does not run despite our efforts to debug the authors’ code.
>
> **W4. Evaluation metrics should also be more comprehensive e.g. results for all 14 lesions.**
>
> **A.** Upon reviewing the comments, we agree that a more comprehensive evaluation with all 14 lesions is appropriate and provides updated results using the 14 lesions above in the revised manuscript. This method of evaluation does not affect the overall results.
>
> **W5. Writing needs to be improved.**
>
> **A.** We have fixed and improved the writing to be clearer to the reader.

---

> > ### Comment · Reviewer_xBeK · 2023-11-23
> > **Thank you for the response**
> >
> > The rebuttal addressed most of my concerns. I will increase my rating to 6. However, I still have concerns about the ethics of using MIMIC data.

---

> ### Author Response · Authors · 2023-11-23
>
> Thank you for helping us improve our work. We appreciate the positive feedback and are glad that our revisions have led to tangible improvements.
>
> In our effort to address the data usage license concerns, we have re-done from scratch the parts of our work that were affected by the MIMIC data usage issue; namely, synthetic question-and-answer generation was re-done using a model that can be run locally (LLaMa2-13B) and therefore without any usage license issues. All evaluation experiments were also re-conducted using the newly trained model. While we anticipate that OpenAI API has already deleted the data (as OpenAI policy stipulates data retention for 30 days only and that the data is not used to train their models nor shared with outside parties [https://openai.com/enterprise-privacy]), we will actively continue to address the data license issues to the best of our ability.
>
> Thank you again for catching this error in the review process.

---

### Official Review · Reviewer_3hh7 · 2023-11-04

**Soundness:** 3 good
**Presentation:** 3 good
**Contribution:** 3 good
**Rating:** 6
**Confidence:** 3

**Summary:**

This study investigates the issue of multi-modal data alignment in large language models, using medical images and reports.

In terms of contributions, the authors propose a bidirectional reasoning generation mechanism that encodes images into tokens, allowing large language models to process and generate both text and images. In model training, a two-stage fine-tuning method is utilized, which not only captures the latent feature distribution of images but also imposes constraints on the model's representation of high-quality samples.

Overall, this paper offers new perspectives on the processing of multi-modal data in large language models, demonstrating their value in the field of medical image processing.

**Strengths:**

1. The author demonstrates the potential of large language models in radiological diagnostics, offering greater flexibility than previous methods in generating diagnostic reports or creating images.

2. Inspired by VQ-GAN, the author ingeniously encodes images into tokens and integrates these image tokens into the large language model (LLM) for fine-tuning, achieving alignment of language and image features in the feature space.

3. The author defines four tasks in this paper, particularly emphasizing the use of the CHATGPT API for Visual Question Answering (VQA), which further strengthens the association between images and text in the feature space. This part of the design is very interesting.

4. During training, the dataset is cleansed, with initial learning of image latent features using low-quality data; the LLM is fine-tuned using the cleansed data.

**Weaknesses:**

1. The experiments in this paper were conducted using only one dataset. As the author mentioned, the quality of the dataset used was limited, necessitating adjustments in the training strategy.

2. The paper lacks visualization of the alignment of images and text in the feature space. Particularly when using VQA for data augmentation, the questions posed by GPT contain rich prior information, focusing the text more on the key lesion areas in the images. I believe that appropriate visualization is necessary to demonstrate the effectiveness of incorporating VQA.

**Questions:**

1. This paper demonstrates the potential application of LLMs in the medical field, but the content of the study extends beyond generating diagnostic reports from radiological images. I hope the author could further clarify in the introduction whether there are specific application scenarios for this research.

2. In '2.1 CLINICAL INFORMATION-PRESERVING CXR TOKENIZATION', the author mentions '...causes loss of clinically important information such as characteristics of microscopic lesions...'. However, in this study, generating image tokens is key to aligning images with text during LLM fine-tuning. Is the mere use of L2 reconstruction loss sufficient to effectively reduce the loss of clinical information? Could there be an enhancement of features in the design of the module network structure? I think this is a very important challenge in this paper, and I hope the difference in experimental results before and after the introduction of L2 reconstruction loss can be explained.

---

> ### Author Response · Authors · 2023-11-20
> **Reply to reviewer 3hh7**
>
> Thank you for your insightful review. We are excited that you agree on the potential of some parts of our approach. Below, we would like to address your concerns and, where applicable, augment our study with further experiments per your comments.
>
> **W1. Experiments are done on just one dataset.**
>
> **A.** Please refer to the general response for added evaluation results on an out-of-domain dataset (IU chest X-ray).
>
> **W2. Visualization of alignment in feature space.**
>
> **A.** While it is true that a direct visualization image and text features in the embedding space would provide the most obvious evidence for image-text alignment, we found a dearth of research on the creation and interpretation of embedding spaces in GPT-based multimodal models. In fact, a recent paper [1] showed that the integration of visual and textual information in text-pretrained multimodal transformer models occurs in the neurons themselves rather than in the feature space. We think that more work on the interpretability of feature spaces of multimodal LLMs is necessary before we are able to apply such techniques to gauge image-text understanding in models. Thus instead, we demonstrate achievement of image-text alignment in the evaluation experiments and ablation studies: training with the VQA task leads to improvements in both the image-based text generation task and the text-based image generation task even though the VQA task does not directly employ these tasks, serving as evidence of improved image-text understanding in the model.
>
> **Q1. Clarification of specific application scenarios.**
>
> **A.** We believe that a model such as ours can be a versatile AI assistant for interacting with large volumes of medical data. For instance, it can be used to find CXR images with specific findings described using natural language from a large dataset of CXRs. This is different from using a traditional image classification network as those networks can only distinguish between predefined labels. This is also different from radiology report generation models that simply take in an image and produce a text output as models that can understand language can provide answers to more contextualized queries, which is more similar to the human radiologist’s workflow (radiologists do not look at the images alone; they understand the clinical context of the patient and the study to produce reports based on them). With confirmation of sufficient accuracy, a model like this can potentially be used to select patients for a clinical trial based on a set inclusion/exclusion criteria or be used for routine, relatively simple clinical tasks such as clearing the absence of certain lesions prior to a procedure.
>
> **Q2. L2 reconstruction loss may not be enough to reduce the loss of clinical information.**
>
> **A.** The preservation of information in the process of tokenizing images into a token space shared by both text and image is definitely an important, major challenge that has to be overcome to develop an accurate model using our methodology. As you have hinted, our employment of the L2 reconstruction loss, while helpful, does not address this problem entirely and there is room for improvement; however, we would like to delve further into why our VQ-GAN training method is able to take a solid step towards that direction:
>
> The original VQ-GAN improves upon the VQ-VAE’s bottleneck architecture and a quantized latent space with LPIPS and adversarial losses. These enhancements improved reconstruction quality, especially with regards to reducing blurry outputs characteristic of models trained with pixel space L1 loss. However, because LPIPS is a loss based on a network trained on natural images (namely Inception-V3), we hypothesized that it is not adequate for extracting features of CXR images and could be improved with a loss tailored for CXR images. Hence, the TXV-loss we use is a variant of the LPIPS loss but calculated using the features of a neural network trained on CXR images (torchxrayvision library). As our ablation study shows that the use of the TXV-loss does increase the performance of the final model, we believe that our hypothesis has at least some merit.
>
> **References**
>
> [1] Schwettmann, Sarah, et al. "Multimodal Neurons in Pretrained Text-Only Transformers." *ICCV*. 2023.

---

> > ### Comment · Reviewer_3hh7 · 2023-11-21
> >
> > Thank you for the author's reply. This is indeed an interesting work.

---

> > > ### Author Response · Authors · 2023-11-22
> > >
> > > > Thank you for the author's reply. This is indeed an interesting work.
> > >
> > > Thank you for your thoughtful consideration of our paper, and we appreciate your positive feedback. We are delighted to hear that you do indeed find our work interesting. Your encouraging words motivate us to continue our research efforts. We are also committed to addressing any further comments or suggestions you may have to enhance the quality of our manuscript.

---

### Author Response · Authors · 2023-11-20
**General Response (1 of 2)**

After receiving the review, we realized an important oversight with regard to our usage of the MIMIC-CXR dataset: ie, we used OpenAI’s ChatGPT API to generate a synthetic question-and-answer dataset using a subset of the text reports in the MIMIC-CXR dataset, but this is, as pointed out by reviewer xBeK, not permitted by the data usage agreement. This was due to our incomplete understanding of the data license, and we will try to responsibly address this error to the best of our ability.

First and foremost, we have removed and re-done the ‘synthetic question-and-answer dataset generation’ part of our work using a model that can be run locally - namely LLaMa2-13B and therefore without any usage license issues. We have subsequently re-conducted all of the evaluation experiments using the newly trained model. Lastly, while we anticipate that the OpenAI API will have already deleted the data (as OpenAI policy stipulates data retention for only 30 days and that the data is not used to train their models nor shared with outside parties [https://openai.com/enterprise-privacy]), we will continue to try to find any post hoc measures that can be taken to mitigate our error. We also sincerely thank reviewer xBeK for catching our mistake.

We briefly review the positive feedback here and go into in-depth discussion in the individual responses below based on the criticisms.

**The strengths of our work include**
1. use of instruction-tuning as a means for achieving image-text alignment in a language-based model (as opposed to a separate neural network - ie, “adapter networks” - that were independently trained for image-text alignments such as CLIP or BLIP-1/2);
2. bidirectional (ie, both text and image) generation from a language-based model based on natural language instruction; and
3. model with a more accurate understanding of chest X-ray images compared to other contemporary models despite utilizing fewer parameters. We show a part of the evaluation results for our updated model (LLM-CXR-Llama2) compared to the previous model (LLM-CXR-ChatGPT) here:

\
***Table 1 (F1). CXR-to-report generation***
|**F1**| **micro** | **macro** | **weighted** |
|:-:|:-|:-:|:-:|
|LLM-CXR-ChatGPT| 0.356| 0.221| 0.349|
|LLM-CXR-Llama2| 0.360| 0.211| 0.350|

\
***Table 3. CXR-VQA***
|Accuracy|All|Presence|Location|Size, severity, type|
|:-:|:-:|:-:|:-:|:-:|
|LLM-CXR-ChatGPT| 56.7%| 60.1%| 49.0%| 53.1%|
|LLM-CXR-Llama2| 44.8%| 41.3%| 50.0%| 62.5%|

\
***Table 4 (F1). Report-to-CXR generation***
|**F1**|**micro**|**macro**|**weighted**|
|:-:|:-:|:-:|:-:|
|LLM-CXR-ChatGPT| 0.780| 0.777| 0.780|
|LLM-CXR-Llama2| 0.806| 0.803| 0.805|

\
Of the three main tasks that our model LLM-CXR can perform (CXR-to-report, report-to-CXR, CXR-VQA), for the first two, new model trained using Llama-13B-generated synthetic VQA was able to reproduce the results of our previous model trained with ChatGPT-generated synthetic VQA and outperform other baselines. In the VQA task, however, the new model shows mildly inferior results than the previous model but remains comparable with other baselines. However, we strongly believe that this result is indicative of the effectiveness of our proposed approach as we were able to replicate the improvements in vision-language alignment with a new independent VQA dataset and also observe that higher quality VQA datasets result in the best results. We are confident that using a more powerful model or better prompting can produce a more-aligned VQA dataset and improve the performance of LLM-CXR further.

In the individual responses below, we delve into our revision efforts.

**Summary of the revisions (more detail in the individual responses):**
1. All results of experiments affected by datasets with the MIMIC-CXR licensing issue were replaced with new results from models free from this issue (thus the numerical results we present will differ slightly from the originally submitted manuscript, but the methodology remains exactly the same)
2. Several other text generation models, including the recently released foundation model RadFM, were added as comparators
3. We now report results for all 14 CheXpert lesions rather than a subset, as suggested by the reviewers, and provide other more extensive evaluations suggested in the reviews

Please note that we are yet to update Appendix E ablation studies using the new model trained on the Llama2-generated dataset due to time constraints but will fully complete this update before publication. As the employed techniques are exactly the same and the newly trained model shows results that are on par with the previous one, we do not anticipate that the results of these ablation studies will differ significantly.

The revised parts of the paper are marked in blue. We welcome additional feedback.

---

> ### Author Response · Authors · 2023-11-22
> **General Response (2 of 2)**
>
> We provide newly available CXR-to-report task results to another evaluation experiment suggested by reviewers: evaluation on the IU Chest X-ray dataset to gauge performance on out-of-distribution images from an external dataset.
>
> | AUROC&uarr; | micro | macro | weighted | Atel. | Cmgl. | Cnsl. | Edema | ECm. | Frac. | Les. |Opac. | NoF. | Eff. | P.O. | Pna. | Pmtx. | Dev |
> |--|--|--|--|--|--|--|--|--|--|--|--|--|--|--|--|--|--|
> | LLM-CXR | **0.795** | **0.538** | **0.593** | 0.553 | 0.594 | 0.497 | 0.548 | 0.497 | 0.508 | 0.502 | 0.556 | 0.616 | 0.601 | 0.499 | 0.532 | 0.498 | 0.529 |
> | XrayGPT | 0.664 | 0.536 | 0.567 | 0.555 | 0.581 | 0.539 | 0.547 | 0.505 | 0.524 | 0.511 | 0.559 | 0.578 | 0.576 | 0.513 | 0.520 | 0.491 | 0.502 |
> | RadFM | 0.724 | 0.520 | 0.542 | 0.515 | 0.543 | 0.502 | 0.519 | 0.489 | 0.510 | 0.512 | 0.508 | 0.554 | 0.584 | 0.531 | 0.503 | 0.497 | 0.518 |
>
> | F1 &uarr; |micro|macro| weighted |Atel.|Cmgl.|Cnsl.|Edema|ECm.|Frac.|Les.|Opac.|NoF.|Eff.|P.O.|Pna.|Pmtx.|Dev.|
> |--|--|--|--|--|--|--|--|--|--|--|--|--|--|--|--|--|--|
> | LLM-CXR | **0.622** | **0.145** | **0.574** | 0.151|0.279|0.0|0.119|0.000|0.029|0.015|0.200|0.811|0.265|0.000|0.079|0.000|0.089|
> | XrayGPT |0.360|0.117|0.430|0.124|0.193|0.045|0.091|0.026|0.059|0.051|0.200|0.602|0.132|0.029|0.058|0.000|0.034|
> | RadFM |0.494|0.113|0.490|0.067|0.157|0.011|0.070|0.000|0.036|0.059|0.097|0.717|0.227|0.059|0.019|0.000|0.061|
>
>
> | Jaccard &uarr; | micro | macro | weighted | unmentioned | possible | negative | positive |
> |-|-|-|-|-|-|-|-|
> | LLM-CXR | **0.823** | **0.344** | **0.840**| 0.900| 0.015| 0.009| 0.452|
> | XrayGPT | 0.443 | 0.219 | 0.558| 0.604| 0.013| 0.038| 0.219|
> | RadFM   | 0.601 | 0.277 | 0.691| 0.743| 0.007| 0.032| 0.328|
>
> The results are similar to that for in-domain data (MIMIC-CXR images): LLM-CXR shows strongest performance, followed by RadFM and then XrayGPT.

---

### Meta-Review · Area_Chair_6Bcs · 2023-12-03

**Metareview:**

This submission receives the following scores: 8, 6, 6, 3, 8. Four out of five reviewers incline to accept the paper, and one reviewer inclines to reject the paper.

The manuscript introduces a novel approach to enhancing Large Language Models (LLMs) with vision-language capabilities, specifically for medical imaging like chest X-rays (CXR). The authors propose an "instruction-finetuning" method using VQ-GAN for image tokenization and integrating these tokens into a pretrained LLM's token space. The fine-tuned model demonstrates proficiency in tasks such as translating CXRs to reports, generating CXRs from reports, and performing CXR-specific visual question answering (VQA), outperforming specialized models in these tasks.

The strengths ("Why Not Lower") and weakness ("Why Not Higher") are provided below. In summary, the paper represents a notable advancement in the integration of vision-language capabilities in LLMs, particularly for medical imaging like CXR. While there are areas for improvement, its strengths in methodology, innovation, and applicability in medical scenarios warrant its acceptance at the conference.

**Justification For Why Not Higher Score:**

- Novelty of Instruction-Finetuning Approach: The distinction between the proposed instruction-finetuning method and previous bidirectional reconstruction tasks in pre-training is subtle, limiting its novelty.
- Motivation and Clinical Application: The paper could more clearly articulate the motivation for the model's training and its specific application in clinical tasks.
- Evaluation Metrics: The evaluation metrics used deviate from standard ones, raising questions about the comparability of the results.
- Lack of Comprehensive Comparison: The paper falls short in providing a comprehensive comparison with non-LLM-based methods in report generation.
- Scalability Concerns: It does not address the scalability of the tokenization and fine-tuning process for larger and more complex medical images, such as 3D CT/MR scans.

**Justification For Why Not Lower Score:**

- Innovative Integration of Visual and Language Features: The method for integrating visual and language features without structural modifications to existing LLMs is innovative and effective.
- Broad Applicability and Performance: The model demonstrates broad applicability in a range of tasks and outperforms specialized models, showcasing its efficacy.
- Contribution to Medical Imaging and LLMs: The paper makes a good contribution to the fields of medical imaging and LLMs, presenting a valuable advancement.
- Solid Methodology and Presentation: The methodology is sound, and the paper is well-presented, making it a good candidate for acceptance.

---

### Decision · Program_Chairs · 2024-01-16

Accept (poster)